# The Elephant in the Room: Towards A Reliable Time-Series Anomaly Detection Benchmark

**Qinghua Liu and John Paparrizos**
Department of Computer Science and Engineering
The Ohio State University
`{liu.11085,paparrizos.1}@osu.edu`

## Abstract

Time-series anomaly detection is a fundamental task across scientific fields and industries. However, the field has long faced the "elephant in the room:" critical issues including flawed datasets, biased evaluation measures, and inconsistent benchmarking practices that have remained largely ignored and unaddressed. We introduce the TSB-AD to systematically tackle these issues in the following three aspects: (i) **Dataset Integrity**: with 1070 high-quality time series from a diverse collection of 40 datasets (doubling the size of the largest collection and four times the number of existing curated datasets), we provide the first large-scale, heterogeneous, meticulously curated dataset that combines the effort of human perception and model interpretation; (ii) **Measure Reliability**: by revealing issues and biases in evaluation measures, we identify the most reliable and accurate measure, namely, VUS-PR for anomaly detection in time series to address concerns from the community; and (iii) **Comprehensive Benchmarking**: with a broad spectrum of 40 detection algorithms, from statistical methods to the latest foundation models, we perform a comprehensive evaluation that includes a thorough hyperparameter tuning and a unified setup for a fair and reproducible comparison. Our findings challenge the conventional wisdom regarding the superiority of advanced neural network architectures, revealing that simpler architectures and statistical methods often yield better performance. The promising performance of neural networks on multivariate cases and foundation models on point anomalies highlights the need for further advancements in these methods. We open-source the benchmark at `https://github.com/TheDatumOrg/TSB-AD` to promote further research.

## 1   Introduction

The explosion of Internet of Things (IoT) applications has significantly increased the volume of sequential measurements [68, 76, 55, 59, 58, 49, 45, 46, 54]. Analytical tasks such as querying [77, 78, 26, 70, 82, 75], forecasting [84, 63, 34], classification [81, 80, 25, 71], and clustering [72, 73, 12, 79, 83] over these ordered sequences of observations, commonly referred to as *time series*, are necessary virtually in every scientific discipline and their corresponding industries [68, 27]. Among these tasks, time-series anomaly detection is widely applied across various sectors [17, 98, 21, 19, 18, 57, 15], ranging from manufacturing quality assurance and data center monitoring to preventing financial fraud. Recently, there has been a surge in interest in this area, primarily driven by advancements in neural network architectures [61, 102, 105, 38] and the availability of diverse datasets [106, 6, 29, 44]. However, the research state of this field has long been plagued by the use of flawed benchmark datasets [106, 104], biased evaluation measures [48, 69, 92], and inconsistent benchmark practices.

The discussion regarding the quality of time-series anomaly detection datasets was initiated by Wu & Keogh [106], who identified common flaws, including triviality, anomaly density, mislabeling, and run-to-failure bias. To address these issues, they introduced a manually curated dataset featuring univariate time series with only a single anomaly, which was often artificially introduced. Therefore,

Table 1: Comparison among TSB-AD and other existing time-series anomaly detection benchmarks. TSB-AD features the most extensive collection and manually curated datasets and provides the broadest coverage of algorithm categories comprising statistical methods (Stat), neural network-based methods (NN), and foundation model-based methods (FM). It also provides a holistic view of evaluation measures with robust hyperparameter tuning across all datasets (HP).

| Benchmark | Dataset | | | | Algorithm | | | Evaluation | |
|---|---|---|---|---|---|---|---|---|---|
| | # Datasets | # Curated TS | Uni | Multi | Stat | NN | FM | HP | # Measures |
| Wu & Keogh [106] | 1 | 250 | ✓ | ✗ | - | - | - | - | - |
| Lai *et al.* [50] | 5 | 0 | ✓ | ✓ | 7 | 2 | 0 | ✗ | 3 |
| Schmidl *et al.* [93] | 15 | 0 | ✓ | ✓ | 49 | 22 | 0 | ✗[*] | 3 |
| Paparrizos *et al.* [74] | 18 | 0 | ✓ | ✗ | 9 | 3 | 0 | ✗ | 9 |
| Wagner *et al.* [104] | 2 | 21 | ✗ | ✓ | 0 | 28 | 0 | ✓ | 3 |
| Zhang *et al.* [112] | 15 | 0 | ✓ | ✓ | 11 | 6 | 0 | ✓ | 4 |
| **TSB-AD (ours)** | 40 | 1070 | ✓ | ✓ | 25 | 10 | 5 | ✓ | 10 |

[*] Hyperparameter tuning is conducted exclusively on a synthetic dataset and applied across the entire evaluation process.

this dataset is not necessarily representative of realistic settings (i.e., virtually all previously published real-world datasets contain more than one anomaly) and disregards other problematic, potentially anomalous regions, leading again to different types of mislabeling problems, as discussed in Section 3.1. Other flaws, such as trivial anomalies, may not necessarily justify the exclusion of a dataset because the real challenge lies in the failure of sophisticated methods to perform effectively, even in these simplified scenarios. Designing a comprehensive benchmark has long been a discussion. However, it seems that "everyone wants to do the model work instead of the data work" [91], resulting in limited new efforts to produce a large-scale, high-quality dataset.

Moreover, the consistent use of flawed evaluation measures continues to create the illusion of progress. For instance, the widely used point-adjustment technique, with the good intention of calibrating the anomaly prediction, favors noisy inputs, and even a random anomaly score has a decent chance of outperforming SOTA methods [48, 92]. Despite the flaw, the measure remains prevalent in recent research of deep-learning-based methods [107, 97, 105, 114], raising concerns about whether the advances are due to improved methods or merely more 'noisy' scoring. With the recently proposed measures [99, 32, 42, 69], it remains unclear which measures to adopt.

Inconsistent benchmarking practices compound these issues. Different communities evaluate their methods on different datasets, which presents a significant challenge when conducting a meta-analysis of their empirical performance. Furthermore, comprehensive benchmark studies may also introduce certain biases. For instance, one of the most cited studies [93] presents an analysis where not all methods were executed on the same datasets (i.e., methods failing to produce results within a specified amount of time or, due to some error, were essentially penalized). In addition, comparing methods with optimized hyperparameters on a single synthetic dataset could lead to misleading conclusions. Moreover, the lack of a unified setup for data preprocessing introduces unfairness in comparison.

From the above, it becomes a necessity to create a robust and reliable benchmark that merges collective wisdom from dozens of published datasets and previous benchmark studies while fixing flaws to facilitate unbiased and consistent benchmark practice. We start with a review of related work (Section 2), then we present our contributions:

- We discuss common flaws in existing datasets and evaluation measures (Section 3).
- We introduce TSB-AD, which comprises 40 datasets—doubling the size of the largest collection, four times the number of existing curated datasets, as well as 40 detection algorithms (Section 4).
- We perform a thorough evaluation on the TSB-AD benchmark to ensure fairness and reliability, and we discuss the insights gained from our research (Section 5).

Finally, we conclude with the implications of our work (Section 6).

## 2 Related Work

### 2.1 Time-Series Anomaly Detection

**[Type of Time Series]** A time series is defined as an ordered sequence of real-valued observations. Consider the signal from $N$ sensors over time $T$, represented as $X = \{x_1, ..., x_T\}$, where $x_t \in \mathbb{R}^N$. A time series is termed *univariate* if $N = 1$ and *multivariate* if $N > 1$.

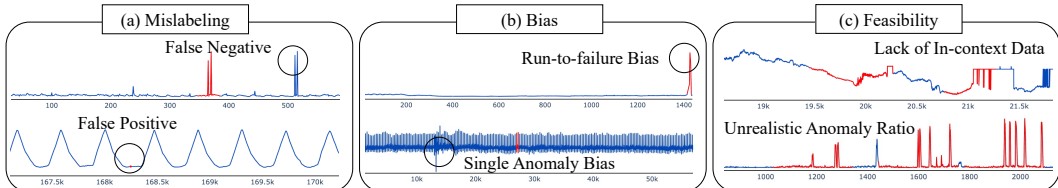

Figure 1: Categorization of common flaws existing in current datasets. Anomalies are marked in red.

**[Type of Anomalies]** Anomalies in time series can occur in the form of a single value or collectively in the form of sub-sequences. Point and contextual anomalies, termed *point-based*, are individual data points that deviate significantly from the majority or expected pattern within a specific context, respectively. Collective anomalies, known as *sequence-based*, consist of sequences of points that deviate from a typical, previously observed pattern.

**[Taxonomy of Detection Algorithms]** The approaches to this task can be categorized based on the level of prior knowledge available: (i) unsupervised, which does not require any labeled data; (ii) semi-supervised, requiring labels only for normal instances; and (iii) supervised, which requires labeled normal and anomalous instances. Due to the limited availability of labeled anomalies, unsupervised or semi-supervised anomaly detection methods are more common. Based on the nature of the processing, the methods can be divided into three categories: distance-based, density-based, and prediction-based methods. Please refer to Appendix B.2 for a more detailed description.

## 2.2 Comparision with Existing Benchmarks

Several benchmark studies have been conducted on time-series anomaly detection, but as previously discussed, they often exhibit certain flaws and biases. We select representative works and compare them with TSB-AD in Table 1. We highlight differences in three key aspects. First, regarding datasets, TSB-AD represents the most extensive collection of time-series anomaly detection datasets to date, nearly doubling the size of the previous largest collection [74]. Beyond the sheer volume, we have meticulously curated the datasets using a principled approach that integrates human perception with algorithmic assistance, details of which are elaborated in Section 4.1. The number of curated time series in TSB-AD is more than four times that of the previous manually curated dataset [106], we further include multivariate time series, and address biases in the UCR [106] dataset. Second, regarding algorithms, TSB-AD is the first to introduce foundation models into anomaly detection benchmarks and encompass representative and top-performing methods identified in earlier studies. Third, in terms of evaluation, our objective is to establish a reliable and frequently updated testbed for fair model performance comparison. We conduct an in-depth investigation into the reliability of evaluation measures and the hyperparameter tuning of various algorithms, providing recommendations based on our findings—aspects that previous studies have neglected [74, 93, 112].

## 3 Common Flaws Creates Illusion of Progress

### 3.1 Flaws in Datasets

We categorize common flaws in existing benchmark datasets, as illustrated in Figure 1.

**[Mislabeling Issues]** Concerns arise over the potential mislabeling of data, which may not be entirely attributable to dataset creators since they had access to additional, non-disclosed data. Our analysis primarily stems from observations of inconsistent labeling, where similar patterns are differently classified—some as anomalies and others not. Figure 1 (a) demonstrates a case where the second spike, similar to a previously labeled anomaly, remains unlabeled, indicating a false negative. Conversely, the anomaly labeled in the lower diagram lacks distinctive features, suggesting a false positive.

**[Bias in Datasets]** The datasets exhibit biases such as the run-to-failure bias, where anomalies predominantly occur towards the end of the time series, as exemplified by the Yahoo dataset [51]. This bias can skew results in favor of algorithms that predict the final data points as anomalies. Furthermore, some datasets, such as the UCR dataset [106], operate under the assumption that the ideal number of anomalies per dataset is one, leading to the marking of only the most prominent anomaly. However, this is often not reflective of real-world conditions where anomalies of either the

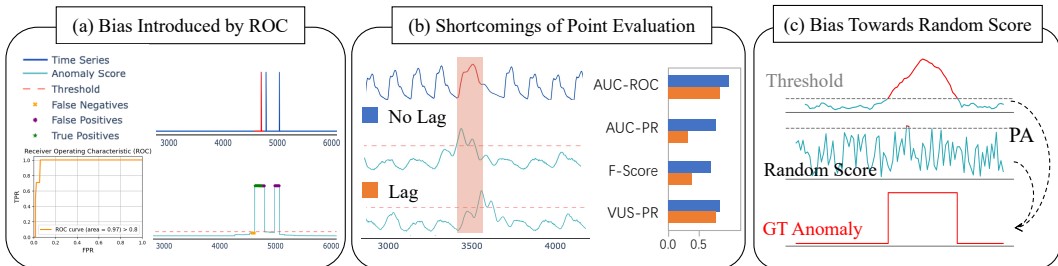

Figure 2: Overview of flaws in evaluation measures.

same or different types will occur multiple times. As depicted in Figure 1 (b), the potential anomalies highlighted within a circle are overlooked.

[Feasibility of Datasets for Anomaly Detection] Often, datasets designed for classification tasks are inappropriately repurposed for anomaly detection by simply reclassifying the minority class as 'anomalous.' However, it is beyond the intended scope of unsupervised anomaly detectors to identify classes with minimal occurrences, thereby constraining their applicability for benchmarking anomaly detection algorithms. Additionally, an unrealistic ratio of anomalies contravenes the fundamental principle that anomalies should be infrequent occurrences as depicted in Figure 1 (c).

## 3.2   Flaws in Evaluation Measures

The detection of anomalies can be viewed as a binary classification problem where data points are classified into normal or abnormal observations. Thus, traditional classification evaluation measures are applicable to anomaly detection. However, the direct application of these measures to time-series anomaly detection causes issues. First, anomaly detection often deals with imbalanced datasets, which can compromise the reliability of certain measures. Second, traditional measures may not account for the sequential nature of time series. For instance, the standard F1 score treats each time step independently, disregarding the temporal dependencies between time steps. Third, while recent advancements have sought to modify these measures to better suit the time-series context, some of these adaptations can still introduce biases, potentially giving misleading indications of progress.

[Unraveling ROC Curve in Anomaly Detection] AUC-ROC [28] evaluates the performance of the model by measuring the area under a curve plotting the true positive rate (TPR) against the false positive rate (FPR), as illustrated in Figure 2 (a). However, anomaly detection typically features a significantly larger count of true negatives compared to false positives, which often yields low FPRs across various thresholds. Consequently, only a small portion of the ROC curve holds relevance under such circumstances. One potential approach to address this issue is to focus solely on specific segments of the curve [11]. In addition, AUC-PR [24] has been advocated as a more informative alternative for imbalanced datasets [60].

Furthermore, previous benchmark studies [93, 74] have assumed that an AUC-ROC value exceeding 0.8 by at least one detection algorithm indicates high-quality labeling. However, as illustrated in Figure 2 (a), despite an AUC-ROC of 0.97, the presence of two false negatives directly challenges this criterion. This scenario not only questions the previous assumption but also underscores the potential for the AUC-ROC measure to overestimate performance in anomaly detection. We argue that relying on a single measure is insufficient for accurately assessing label quality. To address this, we introduce a principled method that combines human perception and algorithmic tests to assess the label quality. A detailed description will be provided in Section 4.1.

[Shortcomings of Point-based Evaluation Measures] The two measures discussed above are point-based evaluation measures in which each point is considered independently, and the detection of each point contributes equally to the AUC. As illustrated in Figure 2 (b), a slight lag in anomaly score results in a significant difference in point-based evaluation measures. However, such lag is often unavoidable due to inconsistencies in labeling practices across datasets and potential delays introduced by anomaly detectors. Consequently, a lack of robustness to lag introduces bias into overall evaluation results. In the context of time series, we argue that two similar anomaly scores with a slight lag should yield approximately the same accuracy measures. For example, a high anomaly score near the boundary of an anomaly should be rewarded similarly to a high anomaly score within the center of

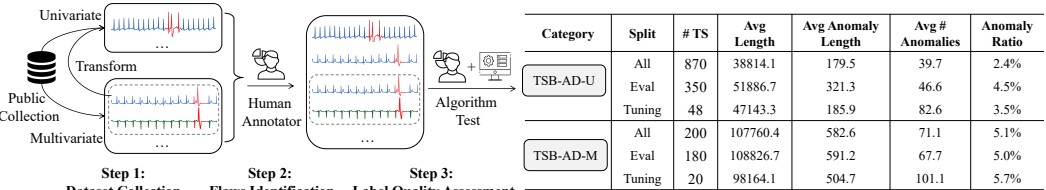

Figure 3: Illustration of dataset construction pipeline and summary statistics of TSB-AD, which comprise 1070 curated time series, with 870 univariate and 200 multivariate.

a range-based anomaly. Range-based evaluation measures, such as VUS-PR [69], have been proposed to accommodate the sequential nature of time series rather than focusing solely on individual points. Further details on other range-based evaluation measures are provided in Section 4.3.

[Bias Towards Random Score] Point Adjustment (PA) assumes that detecting any point within an anomalous segment is considered as if all points within that segment were detected. However, as depicted in Figure 2 (c), this measure tends to favor noisy predictions, whereby even random scores have a decent chance of predicting at least one point in a sequence of ground truth anomalies, performing comparably to state-of-the-art anomaly detectors [48, 104]. Moreover, randomly generated predictions under point adjustment can even outperform SOTA methods, with its point-adjusted F score approaching one as the average length of the anomalies increases. Despite its tendency to substantially overestimate detection performance, this technique remains prevalent in numerous current studies [107, 97, 108, 105, 114]. It is imperative that future evaluations employ unbiased measures to ensure accurate method assessments. Please refer to Section 5.2.1 for further discussion of the reliable evaluation measures.

## 4    TSB-AD: A Reliable Time-Series Anomaly Detection Benchmark

### 4.1    Dataset Overview

#### 4.1.1    Dataset Construction Pipeline

As illustrated in Figure 3, the dataset construction process encompasses three primary steps: (i) dataset collection, which collects datasets introduced over recent decades for anomaly detection in time series; (ii) flaw identification to exclude problematic time series that exhibit common flaws as described in Section 3.1; and (iii) label quality assessment to ensure high-quality labeling, details of which are provided in Section 4.1.2. Each step incorporates the consensus of four human annotators.

[Step 1] The process begins with an extensive collection of 13 univariate and 20 multivariate public time-series anomaly detection datasets which will be further detailed in the Appendix B.1. To enhance the diversity and size of dataset collection, we implement a transformation strategy that converts multivariate time series into univariate formats by treating each channel as an independent series. This strategy is based on the following observations: in some multivariate datasets, only a limited number of channels (often just one) provide valuable information for anomaly detection, while other channels contain categorical, binary, or random values. In addition, our correlation analysis, which evaluates the relationship between the anomaly score of each channel and the ground truth anomaly labels, demonstrates that certain channels exhibit a stronger correlation with the ground truth than others. These observations helped us transform the informative channels of multivariate time series into univariate time series datasets while ensuring the ignored channels do not contribute to the detection of anomalies. During this step, we evaluate multiple anomaly detectors across each channel using distinct evaluation measures. For each time series, we record the highest measure across all detectors as the evaluation result for that data. Subsequently, we select the top 40% of time series for each measure based on these evaluation results (it is important to note that the dataset pruning process is iterative; any time series with suboptimal labeling that passes initial stages can be addressed and removed in subsequent iterations). We then identify the intersection of these selected sets, resulting in an additional 13 univariate time series datasets. By this step, we have obtained a total collection of 46 datasets of univariate and multivariate time series.

[Step 2] Given the lack of consensus in a formal definition of what constitutes a time-series anomaly and the lack of context for producing the labels, we rely on provided anomaly labels to assess

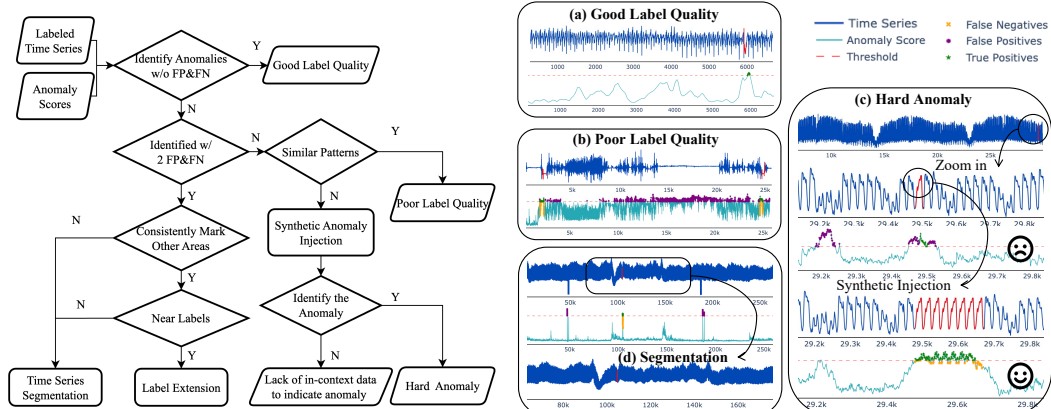

Figure 4: Illustration of the label quality assessment. The flowchart on the left depicts the algorithm testing procedure, starting with an input of labeled time series and the corresponding anomaly scores generated by multiple detectors. The four diagrams on the right exhibit various cases of label quality, including (a) good, (b) poor, (c) hard anomaly, and (d) those that require segmentation.

their suitability for anomaly detection tasks from the perspective of a machine learning practitioner. Moreover, we note that relying solely on measures to assess label quality is not sufficient, as discussed in Section 3.2. Therefore, we further perform manual inspections to identify and remove time series exhibiting common flaws as described in Section 3.1.

**[Step 3]** The task of assessing a dataset's suitability for anomaly detection and the rationality of its labeled anomalies often surpasses human annotators' capabilities. Hence, we utilize anomaly detection algorithms to aid in verifying label quality, as detailed in the following sections.

Upon completing these steps, we have obtained a high-quality set of anomaly-labeled time series, encompassing both univariate (TSB-AD-U) and multivariate (TSB-AD-M) datasets as depicted in Figure 3. The datasets are divided into two partitions: the Eval set, designated for evaluation, and the Tuning set, used for optimizing hyperparameters. However, the number of time series in some univariate datasets, such as UCR [106] and YAHOO [51], is much larger than others. This disparity can lead to a scenario where methods that perform well on that one dataset dominate the entire benchmark. To address this issue, we employ strategic sampling techniques to ensure a more balanced distribution for TSB-AD-U-Eval. Detailed information about the sampling process can be found in Appendix B.1. Please refer to Table 2 for summary statistics of TSB-AD datasets.

### 4.1.2 Label Quality Assessment

As depicted in Figure 4, the objective of the algorithm test is to differentiate among the following scenarios: (i) good label quality, where at least one anomaly detector successfully identifies the anomalies; (ii) bias within the dataset, which can be addressed by segmenting highly confident regions or extending the label; (iii) good label quality but the anomaly is inherently difficult to detect; (iv) the lack of sufficient in-context data to indicate an anomaly, often resulting from the improper application of classification datasets to anomaly detection. Our goal is to establish a benchmark that enables pure anomaly detectors to identify anomalies solely based on the time series data, without relying on external knowledge. Please refer to Appendix B.1.2 for details on algorithm tests.

## 4.2 Time-Series Anomaly Detection Algorithms

In TSB-AD, we have compiled a comprehensive collection of 40 time-series anomaly detection algorithms, comprising statistical, neural network-based, and the latest foundation model-based methods. These methods are selected as representatives of the top-performing models identified by previous benchmark studies [74, 93, 112] and recent research publications. Please refer to the Appendix B.2 for details on the description of algorithms and implementation.

**[Statistical Method]** This category encompasses methods that utilize statistical assumptions to detect anomalies manifesting as deviations from the expected data distribution.

**[Neural Network-based Method]** Methods in this category often adopt a semi-supervised approach, learning to model normal patterns from a history of training data that are anomaly-free and subsequently pinpointing anomalies in new test data.

**[Foundation Model-based Method]** In recent years, there has been a paradigm shift driven by the emergence of foundation models [13]. These models exhibit impressive few-shot or even zero-shot generalization capabilities across a broad spectrum of downstream tasks, often surpassing task-specific models. Works in this area generally fall into two categories: the adaptation of LLMs for time-series anomaly detection tasks, and the utilization of foundation models pre-trained on large-scale time-series data for various time-series applications. In the former category, OFA [114] finetunes the pre-trained GPT backbone on time series data. In the latter category, MOMENT [38] is a family of time-series foundation models for general-purpose time-series analysis, pre-trained through a masked time-series modeling approach. We assess both the zero-shot (ZS) and fine-tuned (FT) detection capabilities of MOMENT. Additionally, models like Lag-Llama [86], Chronos [8], and TimesFM [23], originally designed for time-series forecasting, are repurposed in our benchmarks to perform anomaly detection. This is achieved by comparing forecast values derived from a sliding context against the actual values. To ensure fairness in comparison, we employ the mean squared error between the predictions and the actual values as the anomaly score.

## 4.3 Evaluation Measures

Aligning with previous benchmark evaluations [93, 74, 37], we treat the threshold setting on the anomaly score as an orthogonal problem to our primary focus of model performance evaluation. Our approach either utilizes measures that summarize performance across all possible thresholds or iterates over these thresholds to identify the optimal setting. Considering time-series anomaly detection as both a binary classification and semantic segmentation task [92], we incorporate both *point-wise* measures, which assess the accuracy of detection of individual anomalies, and *range-wise* measures, which offer a robust evaluation of model performance in the context of time series.

**[Point-wise Measures]** For point-wise anomaly detection, we employ widely used measures such as **AUC-ROC**, **AUC-PR**, and **Standard-F1**. For completeness, we additionally incorporate the imperfect yet widely employed **PA-F1** measure, which applies point adjustment to the prediction. The **Event-based-F1** [32] addresses biases from point-adjustment techniques by treating each anomaly segment as an individual event, contributing only once to either a true positive or a false negative.

**[Range-wise Measures]** By considering the sequential nature of time series data, **R-based-F1** [99] expand upon traditional measures by incorporating factors such as existence reward, overlap reward, and cardinality factor. Moreover, **Affiliation-F1** [42] introduces a novel approach by focusing on the proximity between predicted and actual anomaly sequences, measuring the temporal distance between their occurrences. Traditional measures like AUC-ROC and AUC-PR, which assign equal importance to each detection point, often overlook the nuances of labeling consistency and the impact of time lags on anomaly scores. The Volume Under the Surface (VUS) measures, **VUS-ROC** and **VUS-PR** [69], aim to overcome these issues by incorporating a tolerance buffer around outlier boundaries and adopting continuous values over binary labels, thus enhancing the relevance of anomaly scoring. In addition, **PATE** [33] applies proximity-based weighting around anomaly intervals to calculate the weighted version of the area under the Precision and Recall curve.

## 5 Benchmark Evaluation and Analysis

### 5.1 Experimental Setup

**[Tuning/Evaluation Dataset Spiltting]** For both TSB-AD-U and TSB-AD-M, we allocate 15% of the data from each dataset to construct a hyperparameter tuning set. This selection ensures that the tuning set includes representative time series from each dataset, which helps to mitigate bias associated with tuning based solely on one synthetic dataset [93]. Subsequently, the evaluation and comparison of model performance are conducted on the remaining time series.

**[Hyperparameter Tuning]** To ensure fairness by comparing algorithms under their optimal configurations, we design a search space for each algorithm based on recommendations from its original publication or open-source implementation. For instance, for the LOF method [22], we explore a range of models varying the number of neighbors among {10, 20, 30, 40, 50} and the distance

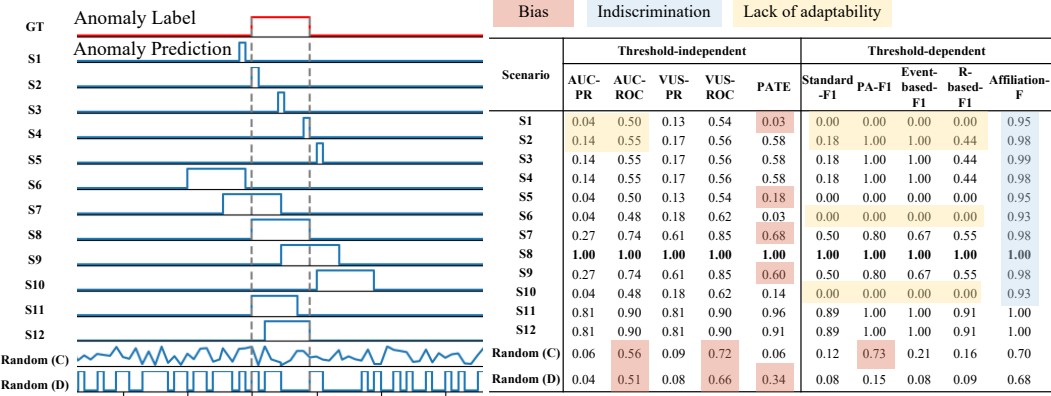

Figure 5: Comparison of evaluation measures for synthetic data examples (left) under different scenarios. S8 corresponds to the oracle situation where prediction is exactly the labeled anomaly. We use different color codings to represent different problematic cases.

| | Threshold-independent | | | | | Threshold-dependent | | | | |
|---|---|---|---|---|---|---|---|---|---|---|
| Scenario | AUC-PR | AUC-ROC | VUS-PR | VUS-ROC | PATE | Standard-F1 | PA-F1 | Event-based-F1 | R-based-F1 | Affiliation-F |
| S1 | 0.04 | 0.50 | 0.13 | 0.54 | 0.03 | 0.00 | 0.00 | 0.00 | 0.00 | 0.95 |
| S2 | 0.14 | 0.55 | 0.17 | 0.56 | 0.58 | 0.18 | 1.00 | 1.00 | 0.44 | 0.98 |
| S3 | 0.14 | 0.55 | 0.17 | 0.56 | 0.58 | 0.18 | 1.00 | 1.00 | 0.44 | 0.99 |
| S4 | 0.14 | 0.55 | 0.17 | 0.56 | 0.58 | 0.18 | 1.00 | 1.00 | 0.44 | 0.98 |
| S5 | 0.04 | 0.50 | 0.13 | 0.54 | 0.18 | 0.00 | 0.00 | 0.00 | 0.00 | 0.95 |
| S6 | 0.04 | 0.48 | 0.18 | 0.62 | 0.03 | 0.00 | 0.00 | 0.00 | 0.00 | 0.93 |
| S7 | 0.27 | 0.74 | 0.61 | 0.85 | 0.68 | 0.50 | 0.80 | 0.67 | 0.55 | 0.98 |
| S8 | 1.00 | 1.00 | 1.00 | 1.00 | 1.00 | 1.00 | 1.00 | 1.00 | 1.00 | 1.00 |
| S9 | 0.27 | 0.74 | 0.61 | 0.85 | 0.60 | 0.50 | 0.80 | 0.67 | 0.55 | 0.98 |
| S10 | 0.04 | 0.48 | 0.18 | 0.62 | 0.14 | 0.00 | 0.00 | 0.00 | 0.00 | 0.93 |
| S11 | 0.81 | 0.90 | 0.81 | 0.90 | 0.96 | 0.89 | 1.00 | 1.00 | 0.91 | 1.00 |
| S12 | 0.81 | 0.90 | 0.81 | 0.90 | 0.91 | 0.89 | 1.00 | 1.00 | 0.91 | 1.00 |
| Random (C) | 0.06 | 0.56 | 0.09 | 0.72 | 0.06 | 0.12 | 0.73 | 0.21 | 0.16 | 0.70 |
| Random (D) | 0.04 | 0.51 | 0.08 | 0.66 | 0.34 | 0.08 | 0.15 | 0.08 | 0.09 | 0.68 |

measures {minkowski, manhattan, euclidean}, resulting in 15 different models. The top-performing model on the tuning set is then selected as the proxy of this detection algorithm for further evaluation. Similarly, for neural network-based methods, we explore hyperparameters such as learning rate and the number of hidden layers [90, 65, 97, 38]. In this way, we obtain over 450 variants from 40 detection algorithms. For details on the candidate hyperparameter settings, please see Appendix C.

## 5.2 Experimental Results and Discussion

To ensure a fair and reliable benchmark evaluation, we begin with the investigation of evaluation measures. Subsequently, we compare various methods through both global and fine-grained analyses. Finally, we compare the insights derived from our study with those from prior benchmark studies to assess progress and consistency in findings. Key results are presented in this section, with supplementary details, including additional comparisons and runtime analyses in Appendix D.

### 5.2.1 Investigation of Evaluation Measures

With 10 evaluation measures available in TSB-AD (Section 4.3), our objective is to identify measures that robustly measure the performance of anomaly detectors without favoring certain anomaly scores or patterns. We explore these measures from the following two perspectives.

[Case Study] To provide a detailed analysis of how each evaluation measure applies its criteria across varying prediction scenarios, we compare different evaluation measures across multiple

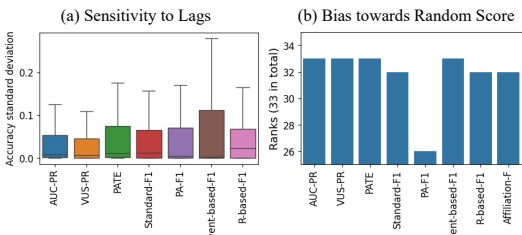

Figure 6: Illustration of reliability of evaluation measures regarding (a) lags and (b) biases.

synthetic examples in Figure 5. Beginning with S1, where predictions occur before the ground truth anomaly, to Random (C), which uses a continuous uniform random score ranging from 0 to 1, and Random (D), a synthetic example with a binary random anomaly score.

We categorize the issues related to evaluation measures into three main categories: (i) bias, referring to measures that favor certain cases or provide inconsistent evaluations under similar conditions; (ii) indiscrimination, where measures fail to meaningfully distinguish between different predictions; and (iii) lack of adaptability, where measures do not account for the specific characteristics of time series data. As shown in Figure 5, with respect to bias, AUC-ROC and VUS-ROC yield high scores for random cases, sometimes exceeding those of S1-S3. PATE and PA-F1 are affected by Random (D) and Random (C), respectively. Specifically, PATE demonstrates sensitivity to lag and inconsistent penalization for early detection, as seen when comparing S1 to S5 and S7 to S9. For discrimination, Affiliation-F provides almost no differentiation across S1 to S10, yielding consistently high scores across scenarios. Regarding lack of adaptability, measures highlighted in yellow fail to account for the time series nature, leading to significant score variations with slight shifts in prediction due to lag.

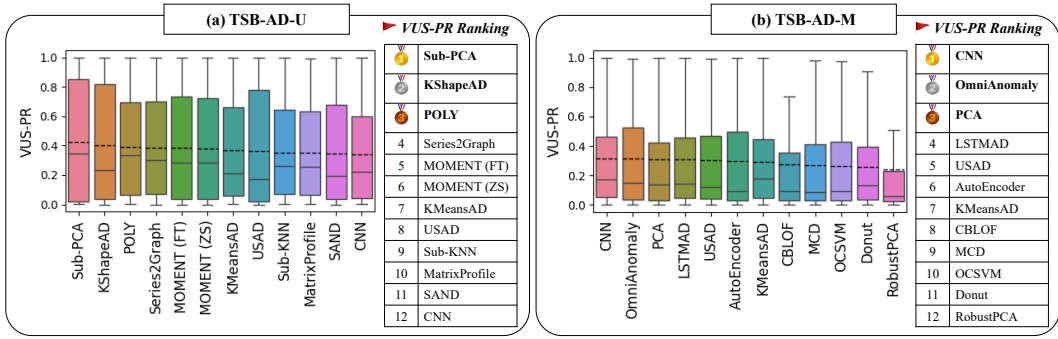

Figure 7: Accuracy evaluation on univariate (a) TSB-AD-U and multivariate (b) TSB-AD-M. In the boxplot, the mean value is marked by a dashed line and the median by a solid line.

**[Quantitative Analysis]** Beyond the case study, we conduct a quantitative analysis to evaluate the sensitivity of evaluation measures to lags—misalignments between predicted anomalies and ground truth— which are inevitable due to inconsistent labeling practices across datasets and potential lags introduced by anomaly detectors. To enhance clarity, we exclude measures with significant bias. Specifically, we introduce lags into the labels and compute the standard deviation of the evaluation results associated with different anomaly scores. As shown in Figure 6 (a), VUS-PR demonstrates substantial robustness compared with other measures. Moreover, as illustrated in Figure 6 (b), when comparing the Random (C) with the anomaly score generated by 32 anomaly detectors, the random score achieves a ranking of 26 under the PA-F1 measure. This finding confirms that PA-F1 exhibits significant bias toward noisy input, rendering it unsuitable for reliable evaluations.

Based on the criteria outlined above, VUS-PR emerges as the most robust (less sensitive to lags), accurate (unbiased and effective across different scenarios), and fair (consistent under similar cases) evaluation measure. In contrast, PATE, while extending the principles of VUS, introduces new challenges that complicate method evaluation and significantly increase computational demands. In the following section, we use VUS-PR as the measure for fair and accurate evaluations. For comprehensive performance analysis, results using additional measures are provided in Appendix D.

### 5.2.2 Benchmark Accuracy Evaluation

Utilizing the most reliable evaluation measures and our curated dataset, we aim to reassess representative anomaly detectors and reveal the current state of research progress through a rigorous benchmarking study. For fair comparison, we apply z-normalization to the time series as a preliminary data preprocessing step, unless an alternative normalization is used in the original implementation. Figure 7 illustrates the evaluation results on TSB-AD-U/M. For clarity, only the top 12 methods are shown here, with a more detailed comparison available in Appendix D. In the boxplot, methods are ordered from left to right according to their rankings based on the average VUS-PR score across all the time series in TSB-AD. While the average VUS-PR value is useful for a global assessment, it should be accompanied by rigorous statistical analysis and CD diagrams to determine whether an improvement in average VUS-PR also reflects an improvement in the average rank per time series. Detailed statistical analysis is provided in Figure 11.

Among the top 12 methods in TSB-AD-U, more than half are statistical approaches, with Sub-PCA dominating the rankings. Only two neural-network-based methods (USAD, CNN) and one foundation-model-based method (MOMENT) are represented, where the fine-tuned version of MOMENT outperforms the zero-shot version. Given that the pretraining datasets of MOMENT include anomaly detection datasets, we provide further analysis of potential data contamination in Appendix D.3. In TSB-AD-M, neural-network-based methods show increased promise, with CNN and OmniAnomaly ranking second and third, respectively. However, statistical methods continue to be highly effective in multivariate cases. Several top-performing methods, including PCA, CNN, and USAD, consistently rank highly across both TSB-AD-U and TSB-AD-M.

### 5.2.3 Analysis on Anomaly Types

We provide a fine-grained analysis of model performance across various anomaly types on TSB-AD-U. Model comparisons are conducted using the Friedman test [31], followed by a posthoc

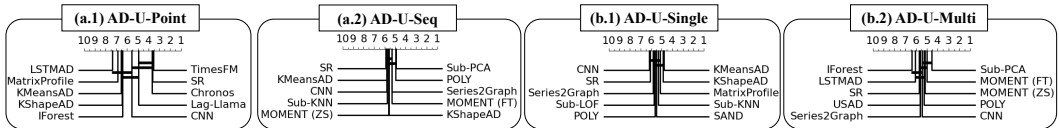

Figure 8: Illustration of model performance on different types of anomaly on TSB-AD-U.

Nemenyi test [66], a widely used statistical approach for comparing multiple algorithms across datasets. Algorithm groups exhibiting no significant performance differences are interconnected horizontally in the Critical Diagram (CD). As depicted in Figure 8, we start by comparing different methods on time series data characterized by point-based anomalies and sequence-based anomalies. Foundation-model-based approaches demonstrate strong potential in detecting point-based anomalies, with TimesFM and Chronos ranking first and third, respectively. However, for sequence-based anomalies, statistical methods continue to dominate, with Sub-KNN and POLY outperforming other methods. Overall, neural networks appear more effective in detecting point anomalies. Subsequently, we analyze the performance difference in scenarios with a single anomaly versus multiple anomalies. No neural-network-based approach ranks as a top candidate in single-anomaly scenarios; however, in more complex cases with multiple anomalies, MOMENT begins to show effectiveness.

### 5.2.4 Discussion

We share the following research insights drawn upon the experimental results on TSB-AD. (i) Statistical-based methods generally demonstrate robust performance, while neural network-based methods do not exhibit the superiority often attributed to them. However, neural networks and foundation models still strive to excel in detecting point anomalies and in handling multivariate cases. (ii) Simpler architectures such as CNNs and LSTMs generally outperform more complex designs, such as advanced transformer architectures. This finding is consistent with recent research [92]. (iii) Foundation models excel at detecting point-based anomalies but struggle with sequence anomalies mostly due to their predictive mechanism, which estimates only one new value per step using a limited look-back window. When faced with long sequence anomalies, the constrained temporal context often leads to reduced performance and noisy scores. The use of flawed point-adjustment techniques that favor these noisy scores further exacerbates this issue, creating an illusion of progress. (iv) The performance of time-series foundation models shows great promise: they not only achieve good performance after fine-tuning but also demonstrate superior zero-shot capabilities when compared to most existing statistical and neural network-based methods. However, a primary concern with foundation models is the risk of data contamination due to the large scale of the pretraining data. Therefore, caution is needed in their deployment. (v) The effort to integrate LLMs into time-series anomaly detection [114] has yielded unsatisfactory results, indicating a significant research gap in this area. (vi) Among the top-performing methods, Sub-PCA and KShapeAD demonstrate exceptional performance, despite having been overlooked as basic baselines for many years and remaining undiscovered in previous extensive evaluation studies [93, 112]. The strong performance of CNN and OmniAnomaly in multivariate cases—contradicting previous benchmarks [93], where KMeansAD was found to be superior—suggests that complex scenarios in multivariate time series require greater modeling capacity, often beyond that of statistical methods.

Finally, it is important to note that no benchmark is perfect, we mainly rely on a limited number of experienced time-series users for manual inspection. We plan to keep expanding the methods and leaderboards and address issues of datasets to ensure a reliable and continuously updated benchmark.

## 6  Conclusion

In this paper, we introduce TSB-AD to address the biases in current benchmarking practices, as well as the issues stemming from flawed datasets and evaluation measures. We provide the first large-scale, manually curated dataset spanning 40 datasets, along with a collection of 40 anomaly detection algorithms and an investigation of 10 evaluation measures. We believe that TSB-AD can serve as a reliable testbed with high-quality datasets (TSB-AD-U/M) and an accurate evaluation measure (VUS-PR), and we advocate for continued efforts in refining dataset creation practices.

## Acknowledgments and Disclosure of Funding

We greatly appreciate Mingyi Huang and Andreas Christian Müller for their valuable feedback and thank other members from The DATUM Lab (i.e., Haojun Li and Fan Yang) for their contributions to the dataset curation process. We also appreciate Purvi Gujarathi and Karthik Jain's preliminary work on collecting datasets and algorithms. Finally, we thank anonymous reviewers for their insightful feedback. This work was supported in part by Meta and Cisco Systems.

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

# Supplementary Material for TSB-AD

*Additional information on algorithms, datasets, and additional experiment settings and results*

## A   Overview

Our supplementary includes the following sections:

- **Section B: More details for TSB-AD.** Datasets and detection algorithms description as well as details for the dataset construction process.
- **Section C: More details for Experiment Setting.** Details for model implementation and hyperparameter setting.
- **Section D: Additional Experiment Results.** Results for additional experiments.

Following NeurIPS Dataset and Benchmark track guidelines, we have shared the following artifacts:

| Artifact | Link | License |
|---|---|---|
| Homepage | `https://thedatumorg.github.io/TSB-AD` | - |
| Github Repository | `https://github.com/TheDatumOrg/TSB-AD` | Apache-2.0 License |

The authors and the DATUM Lab are committed to ensuring its regular upkeep and updates.

Table 2: Summary characteristics of 40 datasets included in TSB-AD. '-' in the 2nd column indicates this dataset is transformed from the multivariate dataset. The 'Category' column indicates whether the datasets feature point anomalies (P) or sequence anomalies (Seq).

| | Name | # TS Collected | # TS Curated | Avg Dim | Avg TS Len | Avg # Anomaly | Avg Anomaly Len | Anomaly Ratio | Category |
|---|---|---|---|---|---|---|---|---|---|
| **TSB-AD-U** | UCR [106] | 250 | 228 | 1 | 67818.7 | 1 | 198.9 | 0.6% | P&Seq |
| | NAB [6] | 58 | 28 | 1 | 5099.7 | 1.6 | 370.1 | 10.6% | Seq |
| | YAHOO [51] | 367 | 259 | 1 | 1560.2 | 5.5 | 2.5 | 0.6% | P&Seq |
| | IOPS [1] | 58 | 17 | 1 | 72792.3 | 25.6 | 48.7 | 1.3% | Seq |
| | MGAB [100] | 10 | 9 | 1 | 97777.8 | 9.7 | 20.0 | 0.2% | Seq |
| | WSD [113] | 210 | 111 | 1 | 17444.5 | 5.1 | 25.4 | 0.6% | Seq |
| | SED [20] | 6 | 3 | 1 | 23332.3 | 14.7 | 64.0 | 4.1% | Seq |
| | TODS [50] | 15 | 15 | 1 | 5000.0 | 97.3 | 18.7 | 6.3% | P&Seq |
| | NEK [96] | 48 | 9 | 1 | 1073.0 | 2.9 | 51.1 | 8.0% | P&Seq |
| | Stock [101] | 90 | 20 | 1 | 15000.0 | 1246.9 | 1.1 | 9.4% | P&Seq |
| | Power [47] | 1 | 1 | 1 | 35040.0 | 4 | 750 | 8.5% | Seq |
| | Daphnet (U) [10] | - | 1 | 1 | 38774.0 | 6 | 384.3 | 5.9% | Seq |
| | CATSv2 (U) [30] | - | 1 | 1 | 300000.0 | 19.0 | 778.9 | 4.9% | Seq |
| | SWaT (U) [62] | - | 1 | 1 | 419919.0 | 27.0 | 1876.0 | 12.1% | Seq |
| | LTDB (U) [35] | - | 9 | 1 | 99700.0 | 127.5 | 144.5 | 18.6% | Seq |
| | TAO (U) [2] | - | 3 | 1 | 10000.0 | 838.7 | 1.1 | 9.4% | P&Seq |
| | Exathlon (U) [44] | - | 32 | 1 | 44075.8 | 3.1 | 1577.3 | 11.0% | Seq |
| | MITDB (U) [35] | - | 8 | 1 | 631250.0 | 68.7 | 451.9 | 4.2% | Seq |
| | MSL (U) [43] | - | 9 | 1 | 3492.0 | 1.3 | 130.0 | 5.8% | Seq |
| | SMAP (U) [43] | - | 19 | 1 | 7700.2 | 1.2 | 210.1 | 2.8% | Seq |
| | SMD (U) [97] | - | 38 | 1 | 24207.7 | 2.4 | 173.7 | 2.0% | Seq |
| | SVDB (U) [39] | - | 20 | 1 | 171380.0 | 36.4 | 292.5 | 3.6% | Seq |
| | OPP (U) [88] | - | 29 | 1 | 16544.8 | 1.4 | 653.4 | 6.4% | Seq |
| **TSB-AD-M** | GHL [29] | 48 | 25 | 19 | 199001.0 | 2.2 | 1035.2 | 1.1% | Seq |
| | Daphnet [10] | 17 | 1 | 9 | 38774.0 | 6.0 | 384.3 | 5.9% | Seq |
| | Exathlon [44] | 72 | 27 | 21 | 60878.4 | 4.3 | 1373.3 | 9.8% | Seq |
| | Genesis [103] | 1 | 1 | 18 | 16220.0 | 3.0 | 16.7 | 0.3& | Seq |
| | OPP [88] | 24 | 8 | 248 | 17426.75 | 1.4 | 394.3 | 4.1% | Seq |
| | SMD [97] | 28 | 22 | 38 | 25466.4 | 8.9 | 112.8 | 3.8% | Seq |
| | SWaT [62] | 4 | 2 | 59 | 207457.5 | 16.5 | 1093.6 | 12.7% | Seq |
| | PSM [3] | 1 | 1 | 25 | 217624.0 | 72.0 | 338.6 | 11.2% | P&Seq |
| | SMAP [43] | 54 | 27 | 25 | 7855.9 | 1.3 | 196.3 | 2.9% | Seq |
| | MSL [43] | 27 | 16 | 55 | 3119.4 | 1.3 | 111.7 | 5.1% | Seq |
| | CreditCard [95] | 1 | 1 | 29 | 284807.0 | 465.0 | 1.1 | 0.2% | P&Seq |
| | GECCO [64] | 1 | 1 | 9 | 138521.0 | 51.0 | 33.8 | 1.2% | Seq |
| | MITDB [35] | 48 | 13 | 2 | 336153.8 | 15.2 | 1846.8 | 2.7% | Seq |
| | SVDB [39] | 78 | 31 | 2 | 207122.6 | 68.3 | 268.2 | 4.8% | Seq |
| | LTDB [35] | 7 | 5 | 2 | 100000.0 | 105.0 | 134.4 | 15.5% | Seq |
| | CATSv2 [30] | 10 | 6 | 17 | 240000.0 | 11.5 | 811.6 | 3.7% | Seq |
| | TAO [2] | 45 | 13 | 3 | 10000.0 | 788.2 | 1.1 | 8.7% | P&Seq |

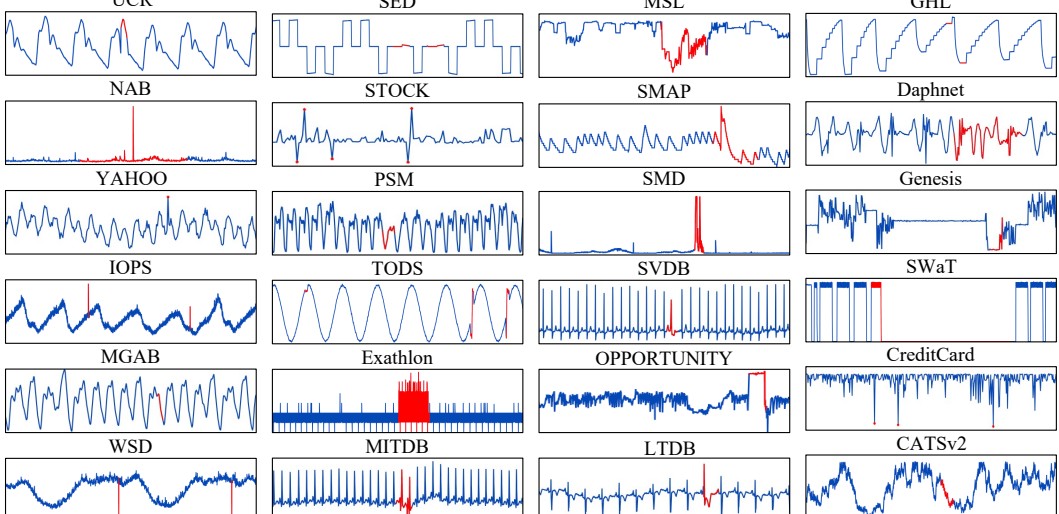

Figure 9: Example time series from TSB-AD, with anomalies highlighted in red. TSB-AD features high-quality labeled time series from a variety of domains, characterized by high variability in length and types of anomalies. Only one channel in a multivariate time series is visualized for brevity.

# B  More Details on TSB-AD Benchmark

In this section, we first provide a brief description of each dataset in our collection and the dataset construction process (Section B.1), and then give an overview of the background related to the time-series anomaly detection algorithms and detectors used in TSB-AD (Section B.2).

## B.1  Dataset Description

### B.1.1  Dataset Collection

We initially identified and collected 13 univariate datasets and 20 multivariate datasets. Following the curation process within TSB-AD, we obtained 23 univariate datasets (including 12 derived from multivariate datasets) and 17 multivariate datasets. Table 2 summarizes relevant characteristics of the datasets, including their size and length, as well as statistics about the anomalies.

TSB-AD includes the following datasets:

- **UCR [106]** is a collection of univariate time series of multiple domains including air temperature, arterial blood pressure, ABP, astronomy, EPG, ECG, gait, power demand, respiration, walking accelerator. Most of the anomalies are introduced artificially.
- **NAB [6]** is composed of labeled real-world and artificial time series including AWS server metrics, online advertisement clicking rates, real-time traffic data, and a collection of Twitter mentions of large publicly-traded companies.
- **YAHOO [51]** is a dataset published by Yahoo labs consisting of real and synthetic time series based on the real production traffic to some of the Yahoo production systems.
- **IOPS [1]** is a dataset with performance indicators that reflect the scale, quality of web services, and health status of a machine.
- **MGAB [100]** is composed of the Mackey-Glass time series, where anomalies exhibit chaotic behavior that is difficult for the human eye to distinguish.
- **WSD [113]** is a web service dataset, which contains real-world KPIs collected from large Internet companies.
- **SED [20]** a simulated engine disk data from the NASA Rotary Dynamics Laboratory representing disk revolutions recorded over several runs (3K rpm speed).
- **Stock [101]** is a stock trading traces dataset, containing one million transaction records throughout the trading hours of a day.
- **TODS [50]** is a synthetic dataset that comprises global, contextual, shapelet, seasonal, and trend anomalies.
- **GHL [29]** contains the status of 3 reservoirs such as the temperature and level. Anomalies indicate changes in max temperature or pump frequency.
- **Daphnet [10]** contains the annotated readings of 3 acceleration sensors at the hip and leg of Parkinson's disease patients that experience freezing of gait (FoG) during walking tasks.
- **Exathlon [44]** is based on real data traces collected from a Spark cluster over 2.5 months. For each of these anomalies, ground truth labels are provided for both the root cause interval and the corresponding effect interval.
- **Genesis [103]** is a portable pick-and-place demonstrator that uses an air tank to supply all the gripping and storage units.
- **OPPORTUNITY [88]** (OPP) is devised to benchmark human activity recognition algorithms (e.g., classification, automatic data segmentation, sensor fusion, and feature extraction), which comprises the readings of motion sensors recorded while users executed typical daily activities.
- **SMD [97]** is a 5-week-long dataset collected from a large Internet company, which contains 3 groups of entities from 28 different machines.
- **SWaT [62]** is a secure water treatment dataset that is collected from 51 sensors and actuators, where the anomalies represent abnormal behaviors under attack scenarios.
- **WADI [7]** is a water distribution dataset with data collected from 123 sensors and actuators under 16 days of continuous operation.
- **SMAP [43]** is real spacecraft telemetry data with anomalies from Soil Moisture Active Passive satellite. It contains time series with one feature representing a sensor measurement, while the rest represent binary encoded commands.
- **MSL [43]** is collected from Curiosity Rover on Mars satellite.
- **CreditCard [95]** is an intrusion detection evaluation dataset, which consists of labeled network flows, including full packet payloads in pcap format, the corresponding profiles, and the labeled flows.

- **GECCO [64]** is a water quality dataset used in a competition for online anomaly detection of drinking water quality.
- **MITDB [35]** contains 48 half-hour excerpts of two-channel ambulatory ECG recordings, obtained from 47 subjects studied by the BIH Arrhythmia Laboratory between 1975 and 1979.
- **SVDB [39]** includes 78 half-hour ECG recordings chosen to supplement the examples of supraventricular arrhythmias in the MIT-BIH Arrhythmia Database.
- **CATSv2 [30]** is the second version of the Controlled Anomalies Time Series (CATS) Dataset, which consists of commands, external stimuli, and telemetry readings of a simulated complex dynamical system with 200 injected anomalies.
- **LTDB [35]** is a collection of 7 long-duration ECG recordings (14 to 22 hours each), with manually reviewed beat annotations.
- **TAO [2]** contains 575, 648 records with 3 attributes which are collected from the Tropical Atmosphere Ocean project.
- **NEK [96]** is collected from real production network equipment.

### B.1.2 More Details on Dataset Construction Process

**[Example Time Series in TSB-AD]** Figure 9 provides representative examples of time series in TSB-AD along with their marked anomalies. For multivariate datasets, only one channel is visualized for brevity. TSB-AD encompasses time series from diverse domains including healthcare, web services, water monitoring, the stock market, and sensor data, representing the most comprehensive collection of high-quality time-series anomaly detection datasets.

**[Sampling Process for TSB-AD-U]** As discussed in Section 4.1.1, some datasets within TSB-AD-U contain significantly more time series than others, which can lead to a scenario where methods that perform well on that one dataset dominate the entire benchmark. To mitigate this and ensure a balanced comparison across various general-purpose anomaly detection algorithms, we implemented a sampling strategy for these datasets, specifically UCR [106], YAHOO [51], and WSD [113]. Given the similarity among time series in the WSD dataset, we randomly chose 20 time series as representative, aligning with the typical dataset size of 20-30 time series in most other collections. From the YAHOO dataset, which originally comprises four subsets, we selected 8 time series from each subset. The UCR dataset comprises time series from multiple domains. To ensure a balanced representation, we sampled 10 time series from each domain category, including air temperature, ECG, and power demand, thereby creating a subset of 70 time series.

**[Description of Algorithm Test]** We detail the methodology used to assess label quality as illustrated in the flowchart in Figure 4. The process begins by evaluating the labeled anomalies within a time series using multiple anomaly detectors. For an anomaly to be considered 'successfully' identified, at least one detector can locate the anomaly within the labeled area (at least one point matches). Success is defined by the existence of a threshold that allows for the identification of the anomaly with no false negatives and false positives, which confirms good label quality. If all anomaly detectors fail to recognize the anomaly, we relax the criteria slightly: if the anomaly is identified with up to two false negatives and false positives, we further investigate the consistency of the anomaly marked by detectors. If marked areas are near the labeled areas, we extend the label to mitigate bias. Conversely, if this is not the case, we segment the time series to preserve areas of high-confidence anomalies. If the anomaly remains unidentified within the adjusted margin of two false negatives and false positives, we assess the time series for similar patterns to the labeled anomaly. The presence of similar patterns indicates a low-quality label and such time series are excluded from the dataset. In cases where similar patterns are absent, we must distinguish between genuinely hard-to-detect anomalies and those cases where there is a lack of in-context data necessary for anomaly detection. Further details on how we differentiate these scenarios are discussed subsequently.

**[Case Study in Algorithm Test]** In scenarios involving hard anomalies where labeled anomalies remain undetected, one approach is to inject synthetic anomalies into the time series to test whether detection algorithms can identify these known anomalies. The most straightforward method involves extending the anomaly label region, as depicted in Figure 4. Observing the response of the anomaly detection algorithms to these adjustments allows us to distinguish between genuinely hard anomalies and cases where the lack of in-context data prevents accurate anomaly identification. The latter often occurs because datasets originally intended for classification are inappropriately utilized for anomaly detection.

## B.2 Time-Series Anomaly Detection Algorithms

### B.2.1 Related Work with More Detail

**[Category of Method]** Based on the nature of the processing, the methods can be divided into three categories: (i) distance-based methods, which analyze subsequences to detect anomalies in time series, primarily by calculating distances to a given model [22, 14]; (ii) density-based methods, identify anomalies by focusing on isolated behaviors within the overall data distribution, rather than measuring nearest-neighbor distances [56, 5]; and (iii) prediction-based methods, which propose to train a model on anomaly-free time series and then reconstruct the data or forecast future points [90, 65]. In this way, the anomalies are identified by significant deviations between predictions and the actual data.

Moreover, the methods can be categorized based on the learning techniques they utilize: (i) statistical methods, which rely on statistical assumptions to identify anomalies as deviations from expected data distributions; (ii) neural network-based methods, which learn to model normal patterns from a history of anomaly-free training data and then identify anomalies in new test data; and (iii) foundation model-based methods, which leverage the knowledge from large models pretrained on extensive time-series or text data for application in time-series anomaly detection tasks. We will organize the algorithms in TSB-AD according to this taxonomy and provide further details in Section B.2.2.

**[Detection Pipeline]** The typical pipeline for time series anomaly detection involves three key stages: (i) preprocessing of time series data, (ii) application of anomaly detection algorithms, and (iii) post-processing of the resulting anomaly scores. In the initial stage, preprocessing may include steps such as normalization or sliding-window transformation, tailored to the specific requirements of the detection algorithm. During the second stage, various anomaly detectors are applied to the processed data to generate scores that reflect the likelihood of each data point being an anomaly, with higher scores indicating a greater probability of abnormality. In the final stage, post-processing usually involves setting a threshold to classify data points as normal or anomalous based on their anomaly scores compared to this predetermined threshold.

### B.2.2 TSB-AD Algorithm List

We organize the detection algorithms in TSB-AD in the following three categories and arrange these algorithms chronologically within each category.

**(i) Statistical Method**

- **(Sub)-MCD [89]** is based on minimum covariance determinant, which seeks to find a subset of all the sequences to estimate the mean and covariance matrix of the subset with minimal determinant. Subsequently, Mahalanobis distance is utilized to calculate the distance from sub-sequences to the mean, which is regarded as the anomaly score.
- **(Sub)-OCSVM [94]** fits the dataset to find the normal data's boundary by maximizing the margin between the origin and the normal samples.
- **(Sub)-LOF [22]** calculates the anomaly score by comparing local density with that of its neighbors.
- **(Sub)-KNN [85]** produces the anomaly score of the input instance as the distance to its $k$-th nearest neighbor.
- **KMeansAD [110]** calculates the anomaly scores for each sub-sequence by measuring the distance to the centroid of its assigned cluster, as determined by the k-means algorithm.
- **CBLOF [41]** is clluster-based LOF, which calculates the anomaly score by first assigning samples to clusters, and then using the distance among clusters as anomaly scores.
- **POLY [52]** detect pointwise anomolies using polynomial approximation. A GARCH method is run on the difference between the approximation and the true value of the dataset to estimate the volatility of each point.
- **(Sub)-IForest [56]** constructs the binary tree, wherein the path length from the root to a node serves as an indicator of anomaly likelihood; shorter paths suggest higher anomaly probability.
- **(Sub)-HBOS [36]** constructs a histogram for the data and uses the inverse of the height of the bin as the anomaly score of the data point.
- **KShapeAD [72, 73, 20]** identifies the normal pattern based on the k-Shape clustering algorithm and computes anomaly scores based on the distance between each sub-sequence and the normal pattern. KShapeAD improves KMeansAD as it relies on a more robust time-series clustering method and corresponds to an offline version of the streaming SAND method [20].

- **MatrixProfile [111]** identifies anomalies by pinpointing the subsequence exhibiting the most substantial nearest neighbor distance.
- **(Sub)-PCA [4]** projects data to a lower-dimensional hyperplane, with significant deviation from this plane indicating potential outliers.
- **RobustPCA [67]** is built upon PCA and identifies anomalies by recovering the principal matrix.
- **EIF [40]** is an extension of the traditional Isolation Forest algorithm, which removes the branching bias using hyperplanes with random slopes.
- **SR [87]** begins by computing the Fourier Transform of the data, followed by the spectral residual of the log amplitude. The Inverse Fourier Transform then maps the sequence back to the time domain, creating a saliency map. The anomaly score is calculated as the relative difference between saliency map values and their moving averages.
- **COPOD [53]** is a copula-based parameter-free detection algorithm, which first constructs an empirical copula, and then uses it to predict tail probabilities of each given data point to determine its level of extremeness.
- **Series2Graph [16]** converts the time series into a directed graph representing the evolution of subsequences in time. The anomalies are detected using the weight and the degree of the nodes and edges respectively.
- **SAND [20]** identifies the normal pattern based on clustering updated through arriving batches (i.e., subsequences) and calculates each point's effective distance to the normal pattern.

### (ii) Neural Network-based Method

- **AutoEncoder [90]** projects data to the lower-dimensional latent space and then reconstruct it through the encoding-decoding phase, where anomalies are typically characterized by evident reconstruction deviations.
- **LSTMAD [61]** utilizes Long Short-Term Memory (LSTM) networks to model the relationship between current and preceding time series data, detecting anomalies through discrepancies between predicted and actual values.
- **Donut [107]** is a Variational AutoEncoder (VAE) based method and preprocesses the time series using the MCMC-based missing data imputation technique.
- **CNN [65]** employ Convolutional Neural Network (CNN) to predict the next time stamp on the defined horizon and then compare the difference with the original value.
- **OmniAnomaly [97]** is a stochastic recurrent neural network, which captures the normal patterns of time series by learning their robust representations with key techniques such as stochastic variable connection and planar normalizing flow, reconstructs input data by the representations, and use the reconstruction probabilities to determine anomalies.
- **USAD [9]** is based on adversely trained autoencoders, and the anomaly score is the combination of discriminator and reconstruction loss.
- **AnomalyTransformer [108]** utilizes the 'Anomaly-Attention' mechanism to compute the association discrepancy.
- **TranAD [102]** is a deep transformer network-based method, which leverages self-conditioning and adversarial training to amplify errors and gain training stability.
- **TimesNet [105]** is a general time series analysis model with applications in forecasting, classification, and anomaly detection. It features TimesBlock, which can discover the multi-periodicity adaptively and extract the complex temporal variations from transformed 2D tensors by a parameter-efficient inception block.
- **FITS [109]** is a lightweight model that operates on the principle that time series can be manipulated through interpolation in the complex frequency domain.

### (iii) Foundation Model-based Method

- **OFA [114]** finetunes pre-trained GPT-2 model on time series data while keeping self-attention and feedforward layers of the residual blocks in the pre-trained language frozen.
- **Lag-Llama [86]** is the first foundation model for univariate probabilistic time series forecasting based on a decoder-only transformer architecture that uses lags as covariates.
- **Chronos [8]** tokenizes time series values using scaling and quantization into a fixed vocabulary and trains the T5 model on these tokenized time series via the cross-entropy loss.
- **TimesFM [23]** is based on pretraining a decoder-style attention model with input patching, using a large time-series corpus comprising both real-world and synthetic datasets.
- **MOMENT [38]** is pre-trained T5 encoder based on a masked time-series modeling approach.

Table 3: Hyperparameter variations of **univariate** detection algorithms. See hyperparameter definitions from TSB-AD (`https://github.com/TheDatumOrg/TSB-AD`). The methods are organized in chronological order within their respective categories.

| | Method | Hyperparameter 1 | Hyperparameter 2 |
|---|---|---|---|
| **Stats** | Sub-MCD [89] | periodicity: [1, 2, 3] | support_fraction: [0.2, 0.4, 0.6, 0.8, None] |
| | Sub-OCSVM [94] | periodicity: [1, 2, 3] | kernel: [linear, poly, rbf, sigmoid] |
| | Sub-LOF [22] | periodicity: [1, 2, 3] | n_neighbors: [10, 20, 30, 40, 50] |
| | LOF [22] | n_neighbors: [10, 20, 30, 40, 50] | metric: [minkowski, manhattan, euclidean] |
| | Sub-KNN [85] | periodicity: [1, 2, 3] | n_neighbors: [10, 20, 30, 40, 50] |
| | KMeansAD [110] | n_clusters: [10, 20, 30, 40] | window_size: [10, 20, 30, 40] |
| | POLY [52] | periodicity: [1, 2, 3] | power: [1, 2, 3, 4] |
| | Sub-IForest [56] | periodicity: [1, 2, 3] | n_estimators: [25, 50, 100, 150, 200] |
| | IForest [56] | n_estimators: [25, 50, 100, 150, 200] | None |
| | Sub-HBOS [36] | periodicity: [1, 2, 3] | n_bins: [5, 10, 20, 30, 40] |
| | MatrixProfile [111] | periodicity: [1, 2, 3] | None |
| | Sub-PCA [4] | periodicity: [1, 2, 3] | n_components: [0.25, 0.5, 0.75, None] |
| | SR [87] | periodicity: [1, 2, 3] | None |
| | Series2Graph [16] | periodicity: [1, 2, 3] | None |
| | KShapeAD [72] | periodicity: [1, 2, 3] | None |
| | SAND [20] | periodicity: [1, 2, 3] | None |
| **NN** | AutoEncoder [90] | win_size: [50, 100, 150] | hidden_neurons: [[64, 32], [32, 16], [128, 64]] |
| | LSTMAD [61] | win_size: [50, 100, 150] | lr: [0.0004, 0.0008] |
| | Donut [107] | win_size: [60, 90, 120] | lr: [0.001, 0.0001, 1e-05] |
| | CNN [65] | win_size: [50, 100, 150] | num_channel, [[32, 32, 40], [16, 32, 64]] |
| | OmniAnomaly [97] | win_size, [5, 50, 100] | lr: [0.002, 0.0002] |
| | TranAD [102] | win_size, [5, 10, 50] | lr, [0.001, 0.0001] |
| | AnomalyTransformer [108] | win_size: [50, 100, 150] | lr: [0.001, 0.0001, 1e-05] |
| | USAD [9] | win_size: [5, 50, 100] | lr: [0.001, 0.0001, 1e-05] |
| | TimesNet [105] | win_size: [32, 96, 192] | lr: [0.001, 0.0001, 1e-05] |
| | FITS [109] | win_size: [100, 200] | lr: [0.001, 0.0001, 1e-05] |
| **FM** | OFA [114] | win_size: [50, 100, 150] | None |
| | Lag-Llama [86] | win_size: [32, 64, 96] | None |
| | Chronos [8] | win_size: [50, 100, 150] | None |
| | TimesFM [23] | win_size: [32, 64, 96] | None |
| | MOMENT [38] | win_size: [64, 128, 256] | None |

Table 4: Hyperparameter variations of **multivariate** detection algorithms. See hyperparameter definitions from TSB-AD (`https://github.com/TheDatumOrg/TSB-AD`). The methods are organized in chronological order within their respective categories.

| | Method | Hyperparameter 1 | Hyperparameter 2 |
|---|---|---|---|
| **Stats** | MCD [89] | support_fraction: [0.2, 0.4, 0.6, 0.8, None] | None |
| | OCSVM [94] | kernel: [linear, poly, rbf, sigmoid] | nu: [0.1, 0.3, 0.5, 0.7] |
| | KNN [85] | n_neighbors: [10, 20, 30, 40, 50] | method: [largest, mean, median] |
| | LOF [22] | n_neighbors: [10, 20, 30, 40, 50] | metric: [minkowski, manhattan, euclidean] |
| | KMeansAD [110] | n_clusters: [10, 20, 30, 40] | window_size: [10, 20, 30, 40] |
| | CBLOF [41] | n_clusters: [4, 8, 16, 32] | alpha: [0.6, 0.7, 0.8, 0.9] |
| | IForest [56] | n_estimators: [25, 50, 100, 150, 200] | max_features: [0.2, 0.4, 0.6, 0.8, 1.0] |
| | HBOS [36] | n_bins: [5, 10, 20, 30, 40] | tol: [0.1, 0.3, 0.5, 0.7] |
| | PCA [4] | n_components: [0.25, 0.5, 0.75, None] | None |
| | RobustPCA [67] | max_iter: [500, 1000, 1500] | None |
| | EIF [40] | n_trees: [25, 50, 100, 200] | None |
| | COPOD [53] | None | None |
| **NN** | AutoEncoder [90] | win_size: [50, 100, 150] | hidden_neurons: [[64, 32], [32, 16], [128, 64]] |
| | LSTMAD [61] | win_size: [50, 100, 150] | lr: [0.0004, 0.0008] |
| | Donut [107] | win_size: [60, 90, 120] | lr: [0.001, 0.0001, 1e-05] |
| | CNN [65] | win_size: [50, 100, 150] | num_channel, [[32, 32, 40], [16, 32, 64]] |
| | OmniAnomaly [97] | win_size, [5, 50, 100] | lr: [0.002, 0.0002] |
| | TranAD [102] | win_size, [5, 10, 50] | lr, [0.001, 0.0001] |
| | AnomalyTransformer [108] | win_size: [50, 100, 150] | lr: [0.001, 0.0001, 1e-05] |
| | USAD [9] | win_size: [5, 50, 100] | lr: [0.001, 0.0001, 1e-05] |
| | TimesNet [105] | win_size: [32, 96, 192] | lr: [0.001, 0.0001, 1e-05] |
| | FITS [109] | win_size: [100, 200] | lr: [0.001, 0.0001, 1e-05] |
| **FM** | OFA [114] | win_size: [50, 100, 150] | None |

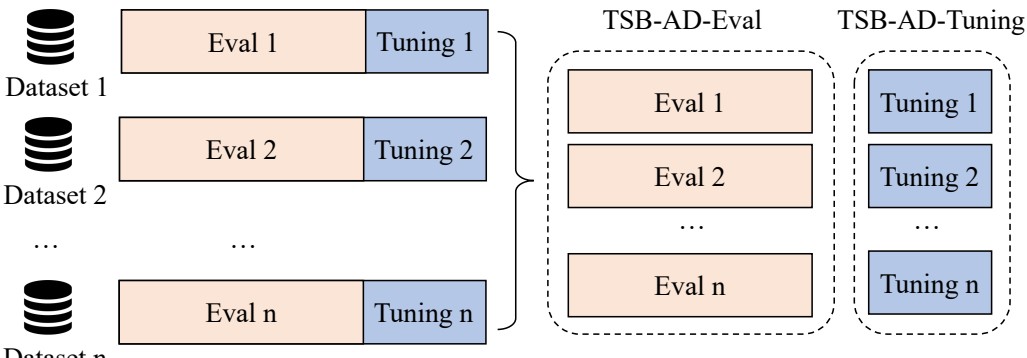

Figure 10: Illustration of TSB-AD-Eval/Tuning construction.

## C   More Details on Experiment Setting

**[Platform]** We conduct our experiments on a server with the following configuration: AMD EPYC 7713 64-Core. The server has two Nvidia A100 GPUs and runs Ubuntu 22.04.3 LTS (64-bit).

**[Implementation]** We have developed a Python library that integrates time-series anomaly detection algorithms in TSB-AD and make it available as an open-source release at `https://github.com/TheDatumOrg/TSB-AD`. This library provides a unified and user-friendly interface, featuring an end-to-end suite of 40 detection algorithms (with ongoing updates planned) and comprehensive evaluation metrics that encompass both point-wise and range-wise assessments.

**[Details on Hyparameter Tuning]** To ensure fairness by comparing algorithms under their optimal configurations, we design a search space for each algorithm based on recommendations from its original publication or open-source implementation. These configurations are detailed in Table 3 for univariate detection algorithms and in Table 4 for multivariate detection algorithms. Please refer to our code base for a more detailed description of each specific hyperparameter.

It is important to note that for univariate detection algorithms, we adapt certain multivariate detection algorithms to univariate contexts to enhance the comprehensiveness. This adaptation involves the use of subsequence versions of multivariate algorithms, such as Sub-MCD [89] and Sub-HBOS [36]. Rather than applying these algorithms directly to individual points of a multivariate time series, we employ sliding window techniques on univariate time series. This approach transforms the univariate data into a pseudo-multivariate format where the dimensionality corresponds to the length of the sliding window. We determine the window length based on the time series periodicity which can be estimated using the autocorrelation function. Given that several time series exhibit multiple periodicities, we consider the max, second, and third max periodicities as three potential options as illustrated in Table 3.

For neural network-based methods, when the original publication does not specify the network architecture, we explore several variants to identify the top-performing model, such as AutoEncoder and LSTMAD. For models with a well-established architecture, our focus shifts primarily to optimizing the learning rate and window length, which is used to segment the time series into batches.

Table 5: Summary accuracy comparison of mean value on TSB-AD-U and TSB-AD-M. The best-performing method as per each metric is marked in **bold** and the second best is marked in underline.

| | Method | AUC-PR | AUC-ROC | VUS-PR | VUS-ROC | Standard-F1 | PA-F1 | Event-based-F1 | R-based-F1 | Affiliation-F1 |
|---|---|---|---|---|---|---|---|---|---|---|
| **TSB-AD-U** | Sub-PCA | **0.37** | 0.71 | **0.42** | 0.76 | **0.42** | 0.56 | 0.49 | **0.41** | 0.85 |
| | KShapeAD | 0.35 | 0.74 | 0.40 | 0.76 | 0.39 | 0.58 | 0.46 | 0.40 | 0.83 |
| | POLY | 0.31 | 0.73 | 0.39 | 0.76 | 0.37 | 0.53 | 0.45 | 0.35 | 0.85 |
| | Series2Graph | 0.33 | 0.76 | 0.39 | 0.80 | 0.38 | 0.65 | 0.50 | 0.35 | 0.85 |
| | MOMENT (FT) | 0.30 | 0.69 | 0.39 | 0.76 | 0.35 | 0.65 | 0.49 | 0.35 | 0.86 |
| | MOMENT (ZS) | 0.30 | 0.68 | 0.38 | 0.75 | 0.35 | 0.61 | 0.49 | 0.36 | 0.86 |
| | KMeansAD | 0.32 | 0.74 | 0.37 | 0.76 | 0.37 | 0.56 | 0.44 | 0.38 | 0.82 |
| | USAD | 0.32 | 0.66 | 0.36 | 0.71 | 0.37 | 0.50 | 0.43 | 0.40 | 0.84 |
| | Sub-KNN | 0.27 | **0.76** | 0.35 | 0.79 | 0.34 | 0.61 | 0.43 | 0.32 | 0.84 |
| | MatrixProfile | 0.26 | 0.73 | 0.35 | 0.76 | 0.33 | 0.63 | 0.44 | 0.32 | 0.84 |
| | SAND | 0.29 | 0.73 | 0.34 | 0.76 | 0.35 | 0.56 | 0.42 | 0.36 | 0.81 |
| | CNN | 0.33 | 0.71 | 0.34 | 0.79 | 0.38 | 0.78 | 0.66 | 0.35 | 0.88 |
| | LSTMAD | 0.31 | 0.68 | 0.33 | 0.76 | 0.37 | 0.71 | 0.59 | 0.34 | 0.86 |
| | SR | 0.32 | 0.74 | 0.32 | **0.81** | 0.38 | **0.87** | 0.67 | 0.35 | 0.89 |
| | TimesFM | 0.28 | 0.67 | 0.30 | 0.74 | 0.34 | 0.84 | 0.63 | 0.34 | **0.89** |
| | IForest | 0.29 | 0.71 | 0.30 | 0.78 | 0.35 | 0.73 | 0.56 | 0.30 | 0.84 |
| | OmniAnomaly | 0.27 | 0.65 | 0.29 | 0.72 | 0.31 | 0.59 | 0.46 | 0.29 | 0.83 |
| | Lag-Llama | 0.25 | 0.65 | 0.27 | 0.72 | 0.30 | 0.77 | 0.59 | 0.31 | 0.88 |
| | Chronos | 0.26 | 0.66 | 0.27 | 0.73 | 0.32 | 0.83 | 0.61 | 0.33 | 0.88 |
| | TimesNet | 0.18 | 0.61 | 0.26 | 0.72 | 0.24 | 0.67 | 0.47 | 0.21 | 0.86 |
| | AutoEncoder | 0.19 | 0.63 | 0.26 | 0.69 | 0.25 | 0.54 | 0.36 | 0.28 | 0.82 |
| | TranAD | 0.20 | 0.57 | 0.26 | 0.68 | 0.25 | 0.58 | 0.43 | 0.25 | 0.83 |
| | FITS | 0.17 | 0.61 | 0.26 | 0.73 | 0.23 | 0.65 | 0.42 | 0.20 | 0.86 |
| | Sub-LOF | 0.16 | 0.68 | 0.25 | 0.73 | 0.24 | 0.57 | 0.35 | 0.25 | 0.82 |
| | OFA | 0.16 | 0.59 | 0.24 | 0.71 | 0.22 | 0.67 | 0.45 | 0.20 | 0.86 |
| | Sub-MCD | 0.15 | 0.67 | 0.24 | 0.72 | 0.23 | 0.54 | 0.32 | 0.24 | 0.81 |
| | Sub-HBOS | 0.18 | 0.61 | 0.23 | 0.67 | 0.23 | 0.60 | 0.35 | 0.27 | 0.79 |
| | Sub-OCSVM | 0.16 | 0.65 | 0.23 | 0.73 | 0.22 | 0.55 | 0.32 | 0.23 | 0.79 |
| | Sub-IForest | 0.16 | 0.63 | 0.22 | 0.72 | 0.22 | 0.63 | 0.34 | 0.23 | 0.80 |
| | Donut | 0.14 | 0.56 | 0.20 | 0.68 | 0.20 | 0.57 | 0.38 | 0.20 | 0.82 |
| | LOF | 0.14 | 0.58 | 0.17 | 0.68 | 0.21 | 0.62 | 0.41 | 0.22 | 0.79 |
| | AnomalyTransformer | 0.08 | 0.50 | 0.12 | 0.56 | 0.12 | 0.53 | 0.34 | 0.14 | 0.77 |
| **TSB-AD-M** | CNN | **0.32** | **0.73** | 0.31 | 0.76 | 0.37 | 0.78 | **0.65** | 0.37 | 0.87 |
| | OmniAnomaly | 0.27 | 0.65 | 0.31 | 0.69 | 0.32 | 0.55 | 0.41 | 0.37 | 0.81 |
| | PCA | 0.31 | 0.70 | 0.31 | 0.74 | 0.37 | 0.79 | 0.59 | 0.29 | 0.85 |
| | LSTMAD | 0.31 | 0.70 | 0.31 | 0.74 | 0.36 | 0.79 | 0.64 | **0.38** | 0.87 |
| | USAD | 0.26 | 0.64 | 0.30 | 0.68 | 0.31 | 0.53 | 0.40 | 0.37 | 0.80 |
| | AutoEncoder | 0.30 | 0.67 | 0.30 | 0.69 | 0.34 | 0.60 | 0.44 | 0.28 | 0.80 |
| | KMeansAD | 0.25 | 0.69 | 0.29 | 0.73 | 0.31 | 0.68 | 0.49 | 0.33 | 0.82 |
| | CBLOF | 0.28 | 0.67 | 0.27 | 0.70 | 0.32 | 0.65 | 0.45 | 0.31 | 0.81 |
| | MCD | 0.27 | 0.65 | 0.27 | 0.69 | 0.33 | 0.46 | 0.33 | 0.20 | 0.76 |
| | OCSVM | 0.23 | 0.61 | 0.26 | 0.67 | 0.28 | 0.48 | 0.41 | 0.30 | 0.80 |
| | Donut | 0.20 | 0.64 | 0.26 | 0.71 | 0.28 | 0.52 | 0.36 | 0.21 | 0.81 |
| | RobustPCA | 0.24 | 0.58 | 0.24 | 0.61 | 0.29 | 0.60 | 0.42 | 0.33 | 0.81 |
| | FITS | 0.15 | 0.58 | 0.21 | 0.66 | 0.22 | 0.72 | 0.32 | 0.16 | 0.81 |
| | OFA | 0.15 | 0.55 | 0.21 | 0.63 | 0.21 | 0.72 | 0.41 | 0.17 | 0.83 |
| | EIF | 0.19 | 0.67 | 0.21 | 0.71 | 0.26 | 0.74 | 0.44 | 0.26 | 0.81 |
| | COPOD | 0.20 | 0.65 | 0.20 | 0.69 | 0.27 | 0.72 | 0.41 | 0.24 | 0.80 |
| | IForest | 0.19 | 0.66 | 0.20 | 0.69 | 0.26 | 0.68 | 0.41 | 0.24 | 0.80 |
| | HBOS | 0.16 | 0.63 | 0.19 | 0.67 | 0.24 | 0.67 | 0.40 | 0.24 | 0.80 |
| | TimesNet | 0.13 | 0.56 | 0.19 | 0.64 | 0.20 | 0.68 | 0.32 | 0.17 | 0.82 |
| | KNN | 0.14 | 0.51 | 0.18 | 0.59 | 0.19 | 0.69 | 0.45 | 0.21 | 0.79 |
| | TranAD | 0.14 | 0.59 | 0.18 | 0.65 | 0.21 | 0.68 | 0.40 | 0.21 | 0.79 |
| | LOF | 0.10 | 0.53 | 0.14 | 0.60 | 0.15 | 0.57 | 0.32 | 0.14 | 0.76 |
| | AnomalyTransformer | 0.07 | 0.52 | 0.12 | 0.57 | 0.12 | 0.53 | 0.33 | 0.14 | 0.74 |

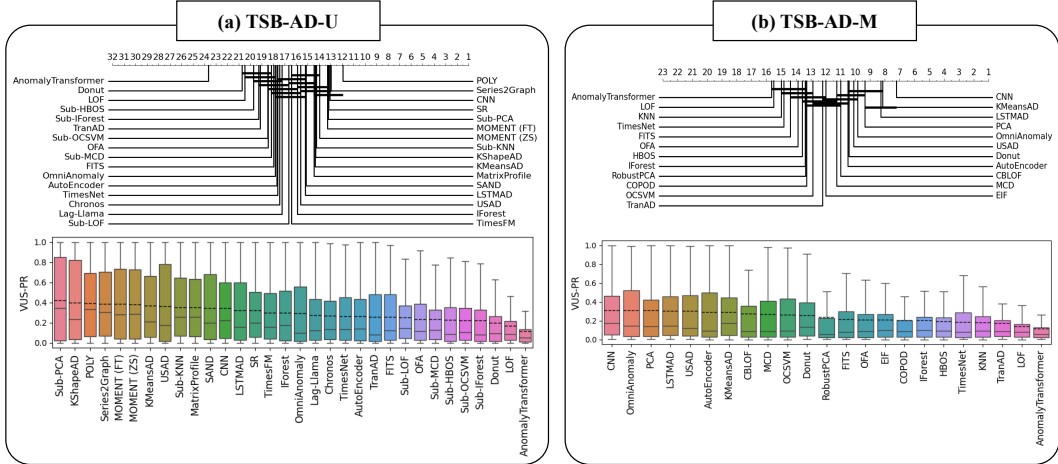

Figure 11: Ranking and score distribution for VUS-PR on (a) TSB-AD-U and (b) TSB-AD-M. The mean value is marked by a dashed line and the median by a solid line in the boxplot.

Table 6: VUS-PR of 32 detection algorithms on 23 datasets of TSB-AD-U. The best-performing method as per each metric is marked in **bold** and the second best is marked in underline.

| Method | CATSv2 | Daphnet | Exathlon | IOPS | LTDB | MGAB | MITDB | MSL | NAB | NEK | OPP | Power | SED | SMAP | SMD | SVDB | SWaT | Stock | TAO | TODS | UCR | WSD | YAHOO |
|---|---|---|---|---|---|---|---|---|---|---|---|---|---|---|---|---|---|---|---|---|---|---|---|
| Sub-PCA | 0.26 | 0.42 | **0.93** | 0.23 | 0.56 | 0.01 | 0.36 | 0.51 | 0.44 | **0.91** | **0.91** | 0.08 | 0.03 | 0.52 | 0.45 | 0.52 | 0.39 | 0.84 | 0.93 | 0.54 | 0.12 | 0.09 | 0.14 |
| KShapeAD | 0.25 | 0.04 | 0.33 | 0.09 | **0.83** | 0.02 | **0.69** | **0.55** | 0.37 | 0.24 | 0.33 | 0.19 | **0.89** | 0.58 | 0.13 | **0.82** | 0.43 | 0.75 | 0.91 | 0.75 | 0.38 | 0.10 | 0.55 |
| POLY | 0.23 | 0.51 | 0.74 | 0.31 | 0.51 | 0.01 | 0.34 | 0.54 | **0.48** | 0.61 | 0.10 | 0.09 | 0.04 | **0.64** | 0.61 | 0.44 | 0.10 | 0.82 | 0.92 | 0.57 | 0.13 | 0.41 | 0.25 |
| Series2Graph | 0.21 | 0.19 | 0.60 | 0.22 | 0.79 | 0.00 | 0.61 | 0.25 | 0.44 | 0.67 | 0.11 | 0.07 | 0.15 | 0.55 | 0.46 | 0.55 | 0.22 | 0.79 | 0.91 | 0.73 | 0.25 | 0.27 | 0.28 |
| MOMENT (FT) | 0.38 | 0.51 | 0.83 | **0.38** | 0.45 | 0.00 | 0.13 | 0.53 | 0.39 | 0.73 | 0.07 | 0.07 | 0.04 | 0.63 | **0.75** | 0.23 | 0.08 | 0.81 | 0.94 | 0.58 | 0.08 | **0.50** | 0.25 |
| MOMENT (ZS) | 0.30 | **0.52** | 0.81 | 0.37 | 0.44 | 0.00 | 0.14 | 0.53 | 0.39 | 0.73 | 0.07 | 0.08 | 0.04 | 0.62 | 0.74 | 0.27 | 0.07 | 0.81 | 0.94 | 0.58 | 0.07 | 0.49 | 0.23 |
| KMeansAD | 0.23 | 0.04 | 0.41 | 0.06 | 0.49 | 0.01 | 0.27 | 0.48 | 0.33 | 0.20 | 0.30 | **0.39** | 0.87 | 0.63 | 0.18 | 0.44 | 0.10 | 0.76 | 0.92 | 0.65 | **0.38** | 0.10 | 0.56 |
| USAD | **0.40** | 0.12 | 0.89 | 0.13 | 0.55 | 0.00 | 0.18 | 0.27 | 0.28 | 0.73 | 0.67 | 0.06 | 0.03 | 0.27 | 0.66 | 0.43 | 0.37 | 0.75 | 0.93 | 0.52 | 0.08 | 0.04 | 0.10 |
| Sub-KNN | 0.29 | 0.04 | 0.47 | 0.10 | 0.58 | 0.24 | 0.36 | 0.33 | 0.29 | 0.23 | 0.30 | 0.21 | 0.87 | 0.51 | 0.14 | 0.56 | 0.10 | 0.75 | 0.92 | 0.65 | 0.37 | 0.10 | 0.31 |
| MatrixProfile | 0.36 | 0.04 | 0.56 | 0.10 | 0.58 | 0.29 | 0.39 | 0.48 | 0.32 | 0.13 | 0.25 | 0.15 | 0.72 | 0.47 | 0.13 | 0.36 | 0.11 | 0.72 | 0.92 | **0.76** | 0.34 | 0.02 | 0.43 |
| SAND | 0.27 | 0.04 | 0.25 | 0.06 | 0.79 | 0.01 | 0.67 | 0.30 | 0.38 | 0.32 | 0.18 | 0.16 | 0.75 | 0.56 | 0.11 | 0.72 | 0.21 | 0.74 | 0.91 | 0.70 | 0.34 | 0.08 | 0.41 |
| CNN | 0.32 | 0.40 | 0.61 | 0.26 | 0.42 | 0.01 | 0.15 | 0.33 | 0.19 | 0.73 | 0.40 | 0.08 | 0.06 | 0.34 | 0.55 | 0.21 | **0.68** | 0.92 | **1.00** | 0.54 | 0.05 | 0.24 | 0.53 |
| LSTMAD | 0.33 | 0.13 | 0.73 | 0.20 | 0.36 | 0.03 | 0.12 | 0.32 | 0.18 | 0.73 | 0.58 | 0.07 | 0.06 | 0.26 | 0.49 | 0.13 | 0.67 | **1.00** | **1.00** | 0.47 | 0.07 | 0.13 | 0.45 |
| SR | 0.28 | 0.20 | 0.73 | 0.24 | 0.29 | 0.01 | 0.07 | 0.22 | 0.20 | 0.50 | 0.33 | 0.10 | 0.07 | 0.29 | 0.36 | 0.08 | 0.35 | **1.00** | **1.00** | 0.64 | 0.07 | 0.22 | 0.61 |
| TimesFM | 0.25 | 0.36 | 0.53 | 0.20 | 0.27 | 0.00 | 0.06 | 0.32 | 0.18 | 0.35 | 0.05 | 0.08 | 0.05 | 0.30 | 0.40 | 0.06 | 0.22 | 0.99 | 0.99 | 0.75 | 0.07 | 0.21 | **0.81** |
| IForest | 0.08 | 0.36 | 0.67 | 0.28 | 0.34 | 0.00 | 0.10 | 0.29 | 0.22 | 0.59 | 0.43 | 0.08 | 0.35 | 0.25 | 0.34 | 0.09 | 0.50 | 0.99 | 0.99 | 0.52 | 0.02 | 0.14 | 0.44 |
| OmniAnomaly | 0.12 | 0.16 | 0.83 | 0.20 | 0.32 | 0.00 | 0.10 | 0.25 | 0.19 | 0.85 | 0.60 | 0.07 | 0.06 | 0.15 | 0.36 | 0.09 | 0.44 | 0.82 | 0.98 | 0.44 | 0.03 | 0.14 | 0.19 |
| Lag-Llama | 0.21 | 0.39 | 0.53 | 0.22 | 0.29 | 0.00 | 0.08 | 0.31 | 0.18 | 0.38 | 0.05 | 0.08 | 0.07 | 0.28 | 0.36 | 0.08 | 0.09 | 0.97 | 0.99 | 0.61 | 0.02 | 0.22 | 0.68 |
| Chronos | 0.10 | 0.31 | 0.45 | 0.18 | 0.26 | 0.00 | 0.06 | 0.18 | 0.18 | 0.34 | 0.06 | 0.08 | 0.06 | 0.19 | 0.32 | 0.06 | 0.14 | 0.99 | 1.00 | 0.70 | 0.07 | 0.18 | 0.80 |
| TimesNet | 0.10 | 0.39 | 0.53 | 0.22 | 0.29 | 0.00 | 0.08 | 0.31 | 0.20 | 0.37 | 0.05 | 0.08 | 0.05 | 0.38 | 0.54 | 0.09 | 0.11 | 0.79 | 0.91 | 0.59 | 0.02 | 0.27 | 0.29 |
| AutoEncoder | 0.18 | 0.09 | 0.36 | 0.25 | 0.69 | 0.01 | 0.07 | 0.27 | 0.32 | 0.51 | 0.12 | 0.09 | 0.41 | 0.49 | 0.14 | 0.32 | 0.38 | 0.72 | 0.93 | 0.65 | 0.08 | 0.04 | 0.10 |
| TranAD | 0.08 | 0.13 | 0.72 | 0.18 | 0.31 | 0.00 | 0.09 | 0.18 | 0.18 | 0.72 | 0.58 | 0.07 | 0.05 | 0.13 | 0.16 | 0.09 | 0.46 | 0.79 | 0.94 | 0.45 | 0.02 | 0.11 | 0.28 |
| FITS | 0.17 | 0.43 | 0.55 | 0.17 | 0.34 | 0.00 | 0.09 | 0.36 | 0.24 | 0.49 | 0.07 | 0.07 | 0.05 | 0.42 | 0.54 | 0.10 | 0.10 | 0.76 | 0.91 | 0.58 | 0.02 | 0.14 | 0.18 |
| Sub-LOF | 0.31 | 0.04 | 0.25 | 0.11 | 0.34 | **0.44** | 0.26 | 0.35 | 0.32 | 0.25 | 0.12 | 0.14 | 0.22 | 0.40 | 0.04 | 0.18 | 0.11 | 0.76 | 0.92 | 0.53 | 0.29 | 0.03 | 0.27 |
| OFA | 0.16 | 0.36 | 0.55 | 0.20 | 0.30 | 0.00 | 0.07 | 0.29 | 0.21 | 0.37 | 0.05 | 0.08 | 0.06 | 0.33 | 0.45 | 0.07 | 0.11 | 0.76 | 0.91 | 0.54 | 0.02 | 0.16 | 0.24 |
| Sub-MCD | 0.37 | 0.04 | 0.23 | 0.13 | 0.24 | 0.01 | 0.11 | 0.16 | 0.19 | 0.11 | 0.32 | 0.30 | 0.12 | 0.30 | 0.08 | 0.07 | 0.09 | 0.75 | 0.90 | 0.64 | 0.26 | 0.15 | 0.28 |
| Sub-HBOS | 0.04 | 0.05 | 0.45 | 0.05 | 0.69 | 0.00 | 0.17 | 0.25 | 0.30 | 0.23 | 0.08 | 0.12 | 0.88 | 0.55 | 0.10 | 0.24 | 0.12 | 0.70 | 0.93 | 0.64 | 0.14 | 0.01 | 0.06 |
| Sub-OCSVM | 0.26 | 0.06 | 0.29 | 0.07 | 0.33 | 0.01 | 0.14 | 0.28 | 0.26 | 0.26 | 0.11 | 0.16 | 0.06 | 0.51 | 0.08 | 0.20 | 0.09 | 0.73 | 0.92 | 0.65 | 0.18 | 0.03 | 0.23 |
| Sub-IForest | 0.05 | 0.07 | 0.49 | 0.04 | 0.66 | 0.00 | 0.10 | 0.24 | 0.36 | 0.30 | 0.22 | 0.07 | 0.12 | 0.79 | 0.47 | 0.09 | 0.27 | 0.13 | 0.69 | 0.90 | 0.66 | 0.10 | 0.06 |
| Donut | 0.08 | 0.06 | 0.45 | 0.10 | 0.31 | 0.00 | 0.10 | 0.20 | 0.18 | 0.47 | 0.18 | 0.09 | 0.14 | 0.31 | 0.29 | 0.08 | 0.47 | 0.78 | 0.91 | 0.48 | 0.01 | 0.06 | 0.12 |
| LOF | 0.06 | 0.14 | 0.21 | 0.12 | 0.26 | 0.00 | 0.06 | 0.15 | 0.16 | 0.38 | 0.15 | 0.09 | 0.11 | 0.15 | 0.13 | 0.05 | 0.12 | 0.75 | 0.91 | 0.49 | 0.02 | 0.09 | 0.37 |
| AnomalyTransformer | 0.05 | 0.07 | 0.13 | 0.06 | 0.27 | 0.00 | 0.09 | 0.14 | 0.14 | 0.23 | 0.07 | 0.09 | 0.09 | 0.09 | 0.18 | 0.07 | 0.10 | 0.75 | 0.90 | 0.46 | 0.01 | 0.02 | 0.07 |

Table 7: VUS-PR of 23 detection algorithms on 17 datasets of TSB-AD-M. The best-performing method as per each metric is marked in **bold** and the second best is marked in underline.

| Method | CATSv2 | CreditCard | Daphnet | Exathlon | GECCO | GHL | Genesis | LTDB | MITDB | MSL | OPP | PSM | SMAP | SMD | SVDB | SWaT | TAO |
|---|---|---|---|---|---|---|---|---|---|---|---|---|---|---|---|---|---|
| CNN | 0.08 | 0.02 | 0.21 | 0.68 | 0.03 | 0.02 | 0.10 | 0.33 | **0.14** | 0.35 | 0.16 | 0.22 | 0.19 | 0.35 | 0.19 | 0.41 | 1.00 |
| OmniAnomaly | 0.04 | 0.02 | 0.34 | 0.84 | 0.02 | **0.07** | 0.00 | **0.44** | 0.11 | 0.22 | 0.18 | 0.16 | 0.12 | 0.17 | **0.35** | 0.15 | 0.81 |
| PCA | 0.12 | **0.10** | 0.13 | **0.95** | **0.20** | 0.01 | 0.02 | 0.24 | 0.07 | 0.15 | **0.30** | 0.24 | **0.36** | 0.11 | 0.45 | 1.00 |
| LSTMAD | 0.04 | 0.02 | 0.31 | 0.82 | 0.02 | 0.06 | 0.04 | 0.30 | 0.09 | 0.22 | 0.17 | 0.24 | 0.16 | 0.33 | 0.15 | 0.16 | 0.99 |
| USAD | 0.04 | 0.02 | **0.34** | 0.84 | 0.02 | 0.06 | 0.00 | 0.41 | 0.12 | 0.23 | 0.18 | 0.19 | 0.11 | 0.16 | 0.32 | 0.15 | 0.81 |
| AutoEncoder | 0.06 | 0.03 | 0.13 | 0.91 | 0.05 | 0.05 | 0.01 | 0.21 | 0.04 | 0.22 | 0.14 | **0.28** | 0.13 | 0.30 | 0.06 | **0.58** | 1.00 |
| KMeansAD | 0.12 | 0.02 | 0.30 | 0.37 | 0.06 | 0.03 | **0.89** | 0.41 | 0.06 | **0.44** | 0.06 | 0.21 | **0.38** | 0.36 | 0.20 | 0.16 | 0.86 |
| CBLOF | 0.06 | 0.03 | 0.10 | 0.86 | 0.03 | 0.02 | 0.02 | 0.20 | 0.04 | 0.21 | 0.14 | 0.19 | 0.14 | 0.22 | 0.07 | 0.29 | 1.00 |
| MCD | **0.13** | 0.06 | 0.14 | 0.80 | 0.03 | 0.01 | 0.06 | 0.21 | 0.04 | 0.23 | 0.17 | 0.26 | 0.10 | 0.26 | 0.07 | 0.54 | 1.00 |
| OCSVM | 0.08 | 0.02 | 0.06 | 0.83 | 0.04 | 0.04 | 0.08 | 0.20 | 0.04 | 0.22 | 0.12 | 0.19 | 0.12 | 0.28 | 0.06 | 0.44 | 0.81 |
| Donut | 0.07 | 0.02 | 0.17 | 0.66 | 0.03 | 0.05 | 0.18 | 0.26 | 0.12 | 0.30 | 0.15 | 0.20 | 0.18 | 0.19 | 0.11 | 0.44 | 0.75 |
| RobustPCA | 0.04 | 0.02 | 0.06 | 0.77 | 0.02 | 0.03 | 0.00 | 0.23 | 0.04 | 0.22 | 0.13 | 0.12 | 0.07 | 0.10 | 0.08 | 0.12 | 1.00 |
| FITS | 0.13 | 0.02 | 0.33 | 0.63 | 0.03 | 0.01 | 0.10 | 0.23 | 0.05 | 0.17 | 0.05 | 0.13 | 0.08 | 0.17 | 0.10 | 0.15 | 0.78 |
| OFA | 0.13 | 0.02 | 0.31 | 0.58 | 0.04 | 0.01 | 0.22 | 0.29 | 0.06 | 0.14 | 0.05 | 0.17 | 0.08 | 0.17 | 0.12 | 0.12 | 0.78 |
| EIF | 0.06 | 0.02 | 0.15 | 0.41 | 0.04 | 0.02 | 0.06 | 0.19 | 0.04 | 0.18 | 0.10 | 0.18 | 0.13 | 0.32 | 0.07 | 0.32 | 0.89 |
| COPOD | 0.05 | 0.05 | 0.13 | 0.40 | 0.04 | 0.03 | 0.08 | 0.21 | 0.04 | 0.21 | 0.12 | 0.10 | 0.19 | 0.07 | 0.31 | 0.31 | 0.99 |
| IForest | 0.05 | 0.03 | 0.13 | 0.35 | 0.04 | 0.05 | 0.08 | 0.21 | 0.04 | 0.21 | 0.18 | 0.19 | 0.09 | 0.26 | 0.07 | 0.39 | 0.93 |
| HBOS | 0.05 | 0.04 | 0.15 | 0.32 | 0.04 | 0.04 | 0.08 | 0.21 | 0.04 | 0.23 | 0.17 | 0.17 | 0.09 | 0.25 | 0.07 | 0.30 | 0.83 |
| TimesNet | 0.07 | 0.02 | 0.27 | 0.42 | 0.03 | 0.01 | 0.02 | 0.27 | 0.07 | 0.17 | 0.06 | 0.14 | 0.09 | 0.14 | 0.11 | 0.14 | 0.79 |
| KNN | 0.07 | 0.02 | 0.25 | 0.33 | 0.11 | 0.01 | 0.04 | 0.19 | 0.04 | 0.18 | 0.06 | 0.12 | 0.12 | 0.30 | 0.06 | 0.11 | 0.78 |
| TranAD | 0.04 | 0.02 | 0.31 | 0.10 | 0.02 | 0.06 | 0.04 | 0.26 | 0.07 | 0.24 | 0.16 | 0.23 | 0.09 | 0.30 | 0.12 | 0.15 | 0.81 |
| LOF | 0.05 | 0.02 | 0.11 | 0.16 | 0.13 | 0.01 | 0.08 | 0.19 | 0.04 | 0.14 | 0.10 | 0.15 | 0.09 | 0.16 | 0.06 | 0.15 | 0.79 |
| AnomalyTransformer | 0.03 | 0.02 | 0.07 | 0.10 | 0.02 | 0.03 | 0.01 | 0.21 | 0.05 | 0.12 | 0.07 | 0.21 | 0.06 | 0.07 | 0.08 | 0.18 | 0.77 |

# D  Additional Experiment Results

In this section, we first present additional benchmark accuracy evaluation results (section D.1) and then provide more detailed analysis of different types of anomalies (Section D.2). Subsequently, we conduct case study on several detection algorithms (Section D.3). Finally, we discuss the runtime results (Section D.4).

## D.1  Addition Results for Benchmark Accuracy Evaluation in Section 5.2.2

In addition to accuracy evaluation in Section 5.2.2, we provide the accuracy comparison of the mean value of all detection algorithms on TSB-AD-U and TSB-AD-M in Table 5. Moreover, we provide the critical diagram and score distribution for VUS-PR (the most reliable and robust evaluation metric as discussed in Section 5.2.1) in Figure 11. In TSB-AD-U, Sub-PCA is identified as the top-performing model under the average VUS-PR value ranking, whereas Sub-POLY excels in the average ranking as demonstrated in CD diagram. In TSB-AD-M, CNN achieves promising results. Furthermore, analysis of model performance under the PA-F1 reveals that neural network-based methods generally achieve higher scores after point adjustment. This observation further underscores the potential bias of this evaluation metric.

Furthermore, apart from the aggregate evaluation results across the entire benchmark, we offer a more detailed accuracy comparison for each dataset, specifically for univariate detection algorithms in Table 6 and for multivariate detection algorithms in Table 7. These comparisons reveal that no single model consistently outperforms others across all datasets. Notably, some methods excel on certain datasets but completely fail on others.

## D.2  Addition Results for Analysis on Anomaly Types in Section 5.2.3

**[Anomaly Type]** We provide additional analysis for model performance across various types of anomalies within TSB-AD-U in Figure 12. For time series characterized by point anomalies, foundation models TimesFM and Choronos emerge as the top-performing models, closely followed by SR. Conversely, for time series with sequence anomalies, POLY ranks highest, succeeded by other statistical methods including Sub-PCA and Series2Graph. In cases with a single anomaly, MatrixProfile and KMeansAD show strong performance, while MOMENT (FT) demonstrates potential when multiple anomalies are present.

**[Anomaly Ratio]** We further investigate the influence of anomaly ratio on model performance in TSB-AD-U (Figure 13) and TSB-AD-M (Figure 14). The analysis reveals that models such as IForest, CNN, and PCA exhibit greater sensitivity to changes in anomaly ratio. Conversely, top-performing methods like KshapeAD demonstrate greater stability across varying anomaly ratios.

## D.3  Case Study

**[Investigation of Data Contamination in Foundation Models]** Identifying time series that a foundation model has not encountered during pretraining is challenging due to the extensive coverage of publicly available time series in their pretraining data and sometimes ambiguous data source usage. Nevertheless, the dedicated evaluation subset, utilized for assessing the performance of a foundation model as detailed in their original publication, serves as a reliable source of time series that the model has not previously encountered. This allows us to utilize the dedicated evaluation subset for the effective analysis of data contamination problems. For instance, MOMENT [38] provides an example of such a case. Our analysis of the zero-shot (ZS) and fine-tuned (FT) versions of MOMENT—where fine-tuning uses the initial segments of the time series as training data—on TSB-AD-U and the dedicated evaluation subset (denoted as MOMENT-Eval), as illustrated in Figure 15, indicates a significant performance decline on Eval. Instead, statistical/data

|  | MOMENT (ZS) | MOMENT (FT) | KShapeAD |
|---|---|---|---|
| TSB-AD-U | 0.38 | 0.39 | 0.40 |
| MOMENT-Eval | 0.12 | 0.14 | 0.32 |
| *Difference* | *0.26* | *0.25* | *0.08* |

Figure 15: Comparative VUS-PR analysis between MOMENT and NORMA on TSB-AD and the dedicated evaluation subset (MOMENT-Eval).

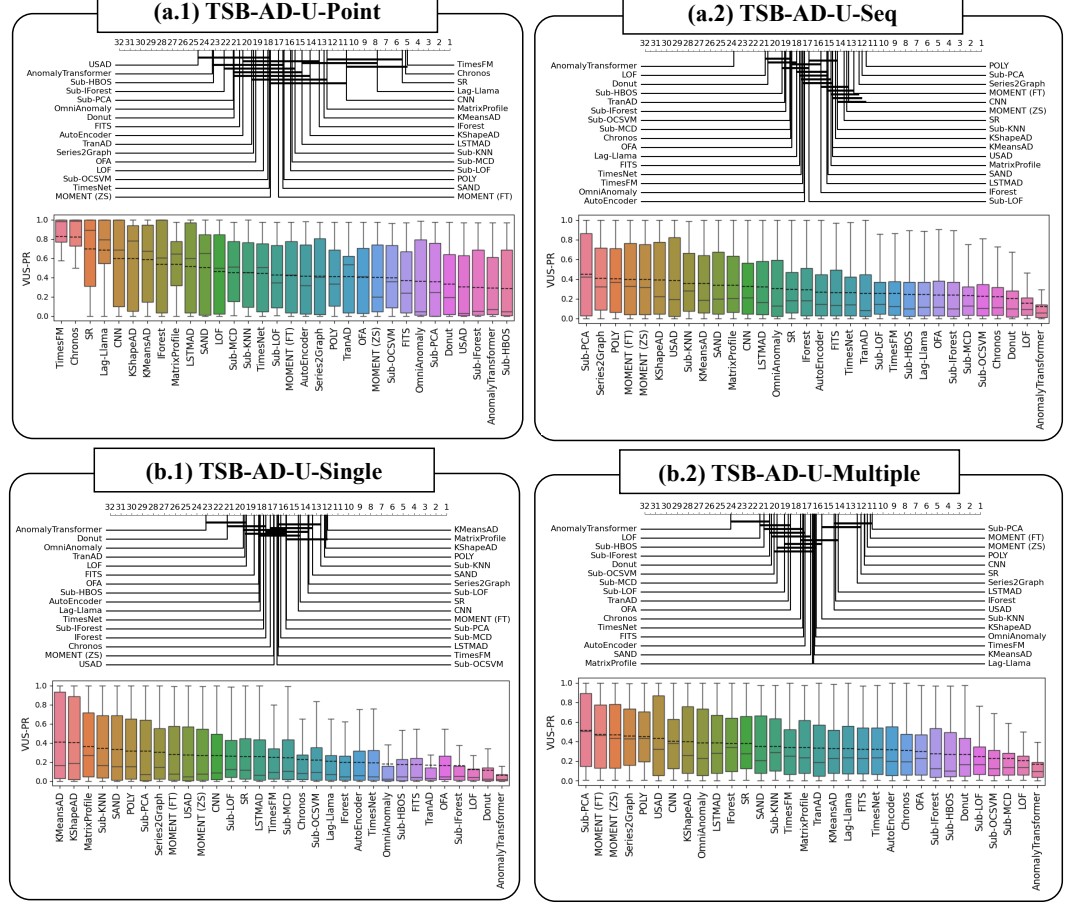

Figure 12: Average ranking and score distribution for VUS-PR on different types of anomaly in TSB-AD-U. The mean value is marked by a dashed line and the median by a solid line in the boxplot.

mining methods (e.g., KShapeAD [72]) do not suffer from such problems and exhibit competitive performance in both settings. This underscores a critical issue of data contamination.

**[Pairwise Comparison]** We perform a pairwise comparison to provide concrete examples to illustrate the disparities in performance. In TSB-AD, we analyze two variants of the MOMENT algorithm. MOMENT (ZS) represents the zero-shot variant, utilizing a pre-trained foundation model directly for anomaly detection tasks. Conversely, MOMENT (FT) involves fine-tuning the pre-trained model on corresponding training data before applying it to anomaly detection. As illustrated in Figure 16, we present a pairwise comparison between MOMENT (FT) and MOMENT (ZS). The results show that MOMENT (FT) significantly outperforms MOMENT (ZS), demonstrating that fine-tuning effectively reduces the distribution gap.

## D.4 Runtime Analysis

During the runtime measurement process, computations are conducted by default on a single CPU process. For neural network-based and foundation models, acceleration is performed using a single GPU. We provide the average runtime of various detection algorithms across the entire benchmark in Figure 17. As anticipated, statistical methods are generally the quickest, followed by neural network-based methods. Foundation models exhibit the slowest runtimes due to their substantial model sizes. Specifically, SR and PCA are the fastest for univariate and multivariate detection, respectively. Notably, while methods such as LOF perform well, they tend to be slower in terms of runtime. In contrast, simpler neural network architectures like CNN and LSTM not only perform effectively but also demonstrate fast runtimes. The deployment of the foundation model may raise concerns regarding its runtime, despite showing promise in terms of detection point anomalies.

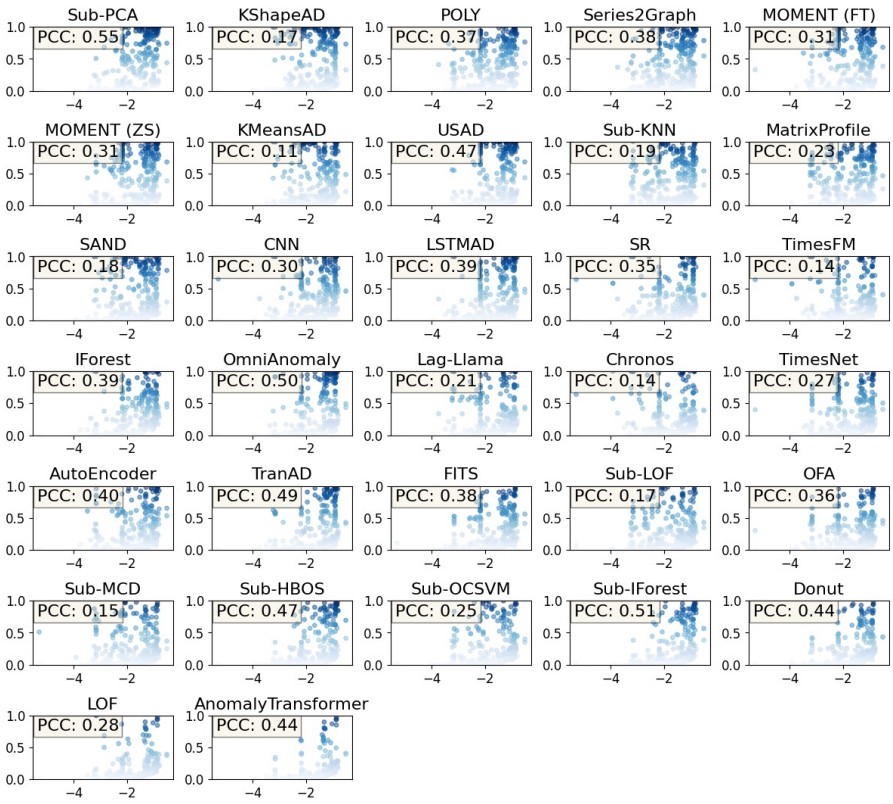

Figure 13: Influence of anomaly ratio on model performance in TSB-AD-U. PCC indicates the Pearson Correlation Coefficient.

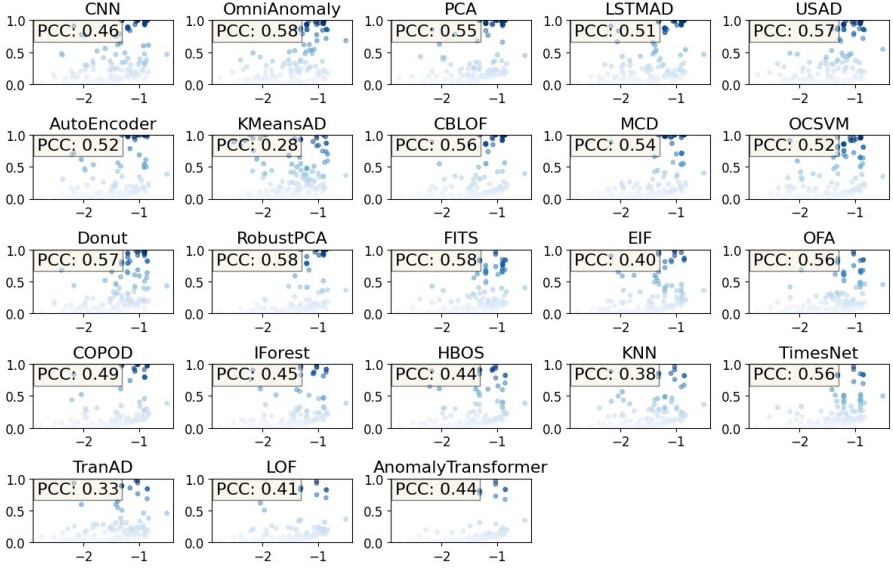

Figure 14: Influence of anomaly ratio on model performance in TSB-AD-M. PCC indicates the Pearson Correlation Coefficient.

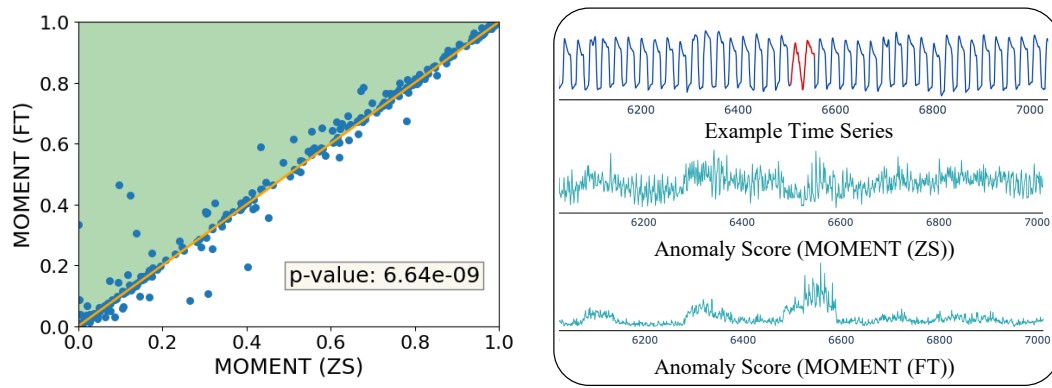

Figure 16: Pairwise comparison between MOMENT (ZS) and MOMENT (FT) across the entire TSB-AD-U under VUS-PR. P-value is determined by the one-sided Wilcoxon signed-rank test.

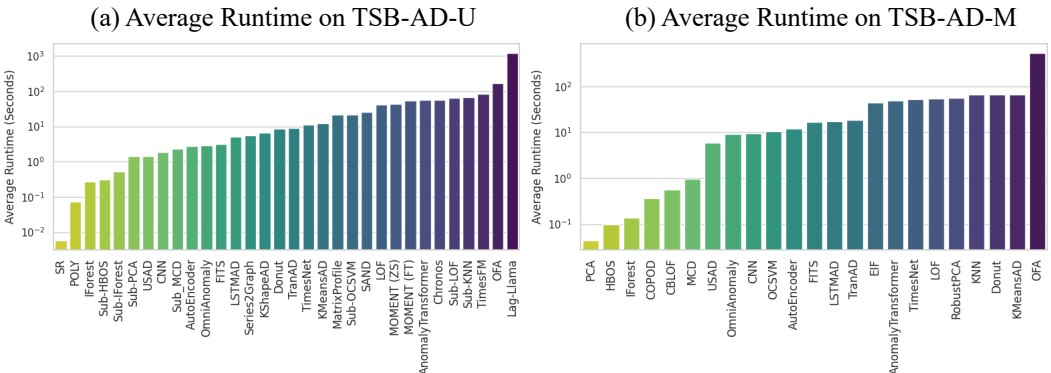

Figure 17: Average runtime on (a) TSB-AD-U and (b) TSB-AD-M.

