# OpenReview forum: "The Elephant in the Room: Towards A Reliable Time-Series Anomaly Detection Benchmark"
_NeurIPS.cc/2024/Datasets_and_Benchmarks_Track — NeurIPS 2024 Track Datasets and Benchmarks Poster_

### Official Review · Reviewer_SW7e · 2024-07-17
**This draft discusses the challenges and solutions in creating reliable benchmarks for time series anomaly detection.**

**Rating:** 4
**Confidence:** 4
**Correctness:** Seem correct.
**Clarity:** The quality of this draft is on average.

**Review:**

Pros.
1. This is a well-written and structured draft. All figures and tables can visually tell me the key information.
2. The technique provided in this draft is straightforward and, in general, correct.
3. The overall presentation effect of the draft is worthy of recognition, and it explains the problem, motivation, method, and experimental results more clearly.
4. It is an important hot topic in the field of time series anomaly detection.

Cons.
1. It seems that lots of previous work has designed benchmarks for different datasets and algorithms; what are your work's advantages and contributions?
2. Are your conclusions still valid when faced with a dataset that has missing values?
3. Most of the experiments and previous studies are repetitive; researchers can draw the same conclusion directly from the past studies without experiments. And similar conclusions have already been presented.
4. Please check the article further for language problems.
5. Please provide more experimental setups, such as the parameters of each model.
6. I don't really think the elephant in the room is relevant to the research topic.

**Strengths:**

See the review part.

**Additional Feedback:**

-

**Documentation:**

Yes

**Limitations:**

yes

**Opportunities For Improvement:**

See the review part.

**Relation To Prior Work:**

Not really. There's a lot of repetition.

**Summary And Contributions:**

This draft proposes that the field of time series anomaly detection faces problems such as dataset deficiencies, biased evaluation metrics, and inconsistent benchmarking practices, and it presents TSB-AD, which systematically addresses these problems by providing large-scale, high-quality datasets, reliable evaluation metrics, and a wide range of detection algorithms, and by open-sourcing the datasets and implementations to facilitate further research. It has been found that simple architectures and statistical methods typically perform better than advanced neural networks, but the underlying models need to be cautious in terms of data contamination.

---

> ### Author Rebuttal · Authors · 2024-08-16
>
> We appreciate the analysis provided for our manuscript. Motivated by the feedback, we are committed to addressing the raised concerns and clarifying any ambiguities as suggested. Please find our response to your concerns below.
>
> > (1) The contributions of our work and the relevance of the elephant in the room to the research topic.
>
> We apologize for any ambiguity in conveying the importance of what we refer to as 'the elephant in the room' within the research topic. This term denotes significant yet overlooked issues, such as flawed datasets, biased evaluation metrics, and inconsistent benchmarking practices. For instance, despite known flaws in evaluation techniques like the point-adjustment method and the quality of datasets, the community still keeps utilizing these problematic resources. Inconsistent benchmarking practices, such as problematic parameter tuning and biased execution rates, further create an illusion of progress.
>
> Moreover, criticisms of prevailing issues are often scattered across different papers and communities, making it difficult for practitioners to gain a holistic understanding of the field's direction. For instance, Wu and Keogh [1] first recognize flaws within the datasets; however, they do not investigate the reliability of the evaluation metrics employed. Similarly, Paparrizos et al. [2] develop and evaluate reliable and robust evaluation metrics, yet point-adjustment method are not included in the analysis. It is essential to acknowledge that the community cannot afford to lose another decade to these misconceptions, and it is imperative to consolidate these scattered insights into coherent practice under one benchmark.
>
> Therefore, our work aims to establish a **reliable** benchmark, built upon the accumulated knowledge of the field, with 3 primary contributions: (i) the first large-scale manually annotated dataset, four times the size of the previously largest curated dataset and twice as large as the largest existing collection; (ii) a rigorous examination of the bias and reliability of current evaluation metrics; (iii) a comprehensive evaluation of detection algorithms including union of the best solutions from prior studies and the first integration of foundation models in this domain.
>
> > (2) The problem with missing values.
>
> Handling time series with missing values poses a challenge that is orthogonal to the task of anomaly detection. Specific tasks, like time-series imputation, are designed to address scenarios involving missing values. Previous research on time-series anomaly detection typically assumes that missing values have been resolved in the preprocessing phase.
>
> > (3) Comparison with other benchmark studies.
>
> We provide comparisons with previous works in the following 2 aspects:
>
> * (i) Direct comparison of evaluation results
>
> |        | NAB  | NAB* | IOPS | IOPS* |
> |--------|------|------|------|-------|
> | NORMA [3]  | **0.67** | 0.58 | 0.55 | **0.76**  |
> | PCA [4]   | 0.63 | **0.69** | **0.77** | 0.74  |
>
> **Table 1:** The comparison of AUC-ROC results obtained from TSB-AD and other papers. \* indicates results from other paper [3].
>
> By analyzing the AUC-ROC results from our TSB-AD and those reported in other studies (as shown in **Table 1**), we identify significant variations in the performance of different methodologies. Specifically, PCA outperforms NORMA on the uncurated NAB dataset, whereas NORMA is more effective on the curated version. Conversely, NORMA excels on the IOPS* dataset but underperforms on the IOPS dataset. These differences in relative rankings could lead to varying assessments of performance and conclusions, which we will discuss further in the following paragraph.
>
> * (ii) Comparison of conclusions drawn from TSB-AD with other works
>
> In comparing conclusions drawn from TSB-AD with those from previous works, we observe discrepancies in assessments of top-performing models. For instance, while NORMA is identified as the top-performing model in our benchmark, it does not receive the same recognition in a previous benchmark [5], where Sub-LOF was considered top-performing. We also identify promising methodologies such as MCD, which have previously been overlooked. Additionally, we uncover new insights, including the performance of foundation models in time series analysis—the first in benchmark studies—and the effectiveness of DNN-based methods under sequence anomalies. We believe that deriving conclusions from a reliable testbed is more convincing and beneficial for advancing the community.
>
> Moreover, to enhance the reliability of those findings, given that current reports in various papers may have been based on different subsets of the datasets under different evaluation metrics, we will conduct further analyses on both pre- and post-curated datasets within TSB-AD. This approach will ensure a more fair and accurate comparison to understand how concerning are previously reported results in the community. We will provide a more detailed comparative analysis in the revised manuscript.
>
> > (4) Language problem.
>
> We will review the entire manuscript to address any linguistic inaccuracies and enhance readability.
>
> > (5) Details on experimental setups.
>
> Due to space limitations in the main text, we have provided a detailed description of the experimental settings, including the parameters for each model in TSB-AD, in Appendix C.
>
> -------
> **Reference:**
>
> [1] R. Wu and E. J. Keogh. Current time series anomaly detection benchmarks are flawed and are creating the illusion of progress. TKDE 2021.
>
> [2] J. Paparrizos et al. Volume under the surface: a new accuracy evaluation measure for time-series anomaly detection. VLDB 2022.
>
> [3] P. Boniol et al. Unsupervised and scalable subsequence anomaly detection in large data series. VLDB 2021.
>
> [4] C. C. Aggarwal. Outlier Analysis. Springer International Publishing, 2 edition, 2017.
>
> [5] S. Schmidl et al. Anomaly detection in time series: a comprehensive evaluation. VLDB 2022.

---

> > ### Author Rebuttal · Authors · 2024-08-26
> >
> > Dear reviewer,
> >
> > Thank you for your effort during this tight review timeline. We would be grateful if you could confirm that you have read our responses and let us know whether we have successfully addressed your concerns.
> >
> > Considering that the other three reviewers have all praised our work, highlighting that (1) “this submission checks all the boxes of a great contribution to this track;” (2) “the findings challenge conventional wisdom;” and (3) “the paper provides a detailed analysis of the issues with existing time series anomaly detection benchmarks from three perspectives and offers solutions,” we are eager to discuss any points that may remain open and may hold you from increasing your rating.

---

### Official Review · Reviewer_Tcpd · 2024-07-19
**Comprehensive benchmark paper**

**Rating:** 8
**Confidence:** 5

**Review:**

This submission checks all the boxes of a great contribution to this track: Not only does it provide new datasets, it also outlines issues with existing ones, with their evaluation, and it provides a comprehensive evaluation, comprising different types of machine-learning techniques. I am very excited about the prospect of making these datasets available to a wider community, since time-series analysis in machine-learning research suffers from numerous drawbacks, and there are few critical voices. This is in stark contrast to the data-mining community, and I thus believe it is time to bring this perspective into machine learning as well.

The paper manages to outline the issues with existing datasets and metrics, while also providing an invaluable resource in terms of modern benchmarks. It is great to see such a diligent evaluation and construction in practice. My main reservations about this paper concern the way the data is stored and distributed (see below) but I believe that these issues can be addressed during the revision process.

**Strengths:**

The paper is comprehensive, dealing with data, metrics, and models. I find the subsequent aspects to be very strong:

- Discussion of failings of existing datasets.
- Discussion of useful metrics.
- Suite of different models for performance assessment.

Taken together, I am sure this will be a strong contribution to the community.

**Additional Feedback:**

None.

**Clarity:**

The paper is very well written, I only have some issues understanding the definition of anomaly (see comment above).

Some minor typos:

- 'principal method' --> 'principled method'
- 'F score' --> 'F-score'
- 'our study 5.2.1' --> 'our study (Sec. 5.2.1)' or something similar

**Correctness:**

Both the paper and the dataset are correct to the best of my understanding. I am not sure whether `CSV` is really the best exchange format for such time-series data, but I believe that for a first version/release of the data, this will be sufficient. I strongly urge the authors to provide a _canonical_ dataset loader, though, since `CSV` parsing can be notoriously unstable, depending on the implementation of the underlying parser.

**Documentation:**

The repository and appendix documentation are quite good. I am missing some aspects of the data collection and maintenance, though. The Google Drive link is currently *not* an appropriate way for hosting the data.

**Ethics:**

All ethical aspects are sufficiently discussed. As far as I can tell, none of the datasets is ethically problematic in any way, since there is no identifying data about passengers, for instance.

**Limitations:**

Limitations are addressed adequately except for the definition of time-series anomalies (as outlined above), which would require some additional discussion.

**Opportunities For Improvement:**

The main opportunities for for improvement concern the following aspects:

- l. 129--...: More details and insights on the shortcomings of measures based on individual points could be provided.
- Section 4.1.1 requires more details. I had a hard time following it since some of the discussion struck me as circular. Please provide more details on what constitutes an anomaly here.
- The dataset is currently neither versioned nor hosted with a persistent URI. Please use a repository like [Zenodo](https://zenodo.org/) for this purpose.
- A better description of the figures, in particular of the critical difference plots in Fig. 7, would be highly warranted. I doubt that many machine-learning researchers are familiar with this type of analysis, and this would be a good opportunity to change that!

**Relation To Prior Work:**

All prior work is clearly discussed and the paper makes a strong, convincing delineation to it.

**Summary And Contributions:**

This submission presents a comprehensive dataset for anomaly detection in time series. Next to presenting new time-series datasets, the submission also points out pitfalls in their evaluation (termed *metric reliability*) and provides a set of benchmarks comprising different types of methods. Not only does the paper provide a highly-relevant source of new datasets, it also manages to challenges some (folklore) assumptions concerning the superiority of neural-network architectures.

---

> ### Author Rebuttal · Authors · 2024-08-16
>
> Thank you for the constructive feedback and the positive aspects noted in your review. We are grateful for the opportunity to enhance our work based on your valuable suggestions.
>
> > (1) More details on the shortcomings of point-based evaluation measures in Section 3.2.
>
> We will expand this discussion in the revised manuscript in Section 3.2 to provide a more thorough critique. Specifically, we will demonstrate how point-based evaluation metrics are less robust to lags and noise in anomaly scores. These vulnerabilities are common due to manufacturing issues, external causes, inconsistent labeling practices across datasets, and potential lags introduced by the anomaly detectors themselves. We contend that evaluating time-series anomaly detectors should account for the temporal characteristics of time-series data, and we advocate against the exclusive reliance on point-based metrics.
>
> > (2) Definition of the time-series anomaly and what constitutes an anomaly.
>
> Given the lack of consensus in a formal definition of what constitutes a time-series anomaly and the lack of context for producing the labels, we rely on provided anomaly labels to assess their suitability for anomaly detection tasks from the perspective of a machine learning practitioner. For example, if one region is labeled as an anomaly while another exhibiting similar patterns is not, this inconsistency suggests a mislabeling issue that necessitates removing the time series to prevent confusion. In essence, our role does not extend to generating anomaly labels but to evaluating existing labels' quality. This approach facilitates the establishment of practical rules for identifying instances that deviate from expected norms. We will enhance the clarity and depth of the discussion for the definition of anomaly in Section 4.1.1 in the revised manuscript.
>
> > (3) The maintenance of datasets.
>
> We will officially release our datasets on Zenodo and a new portal we prepare (timeseries.org) upon acceptance. Additionally, we are in the process of developing a Pypi package index (pip install version of TSB-AD), along with a canonical dataset loader and Croissant format, to facilitate straightforward utilization of our benchmark suite. Looking forward, we are committed to continually updating the benchmark with new emerging datasets and methodologies and will host an online leaderboard on timeseries.org.
>
> > (4) More details on the CD diagram.
>
> Thank you for the suggestion. This presents a good opportunity to highlight the use of statistical analysis to compare multiple algorithms across various datasets. We will include additional details in the revised manuscript.

---

> > ### Author Rebuttal · Authors · 2024-08-26
> >
> > Dear reviewer,
> >
> > We greatly value your efforts during this tight review timeline and are thankful for your acknowledgment of our paper's strong contributions to the field. In our previous response, we have addressed your comments regarding more details on the limitations of point-based evaluation measures, the definition of time-series anomalies, and the maintenance of datasets.
> >
> > We are really happy and respect your current rating for our work. However, based on the acceptance statistics from recent years (https://papercopilot.com/statistics/neurips-statistics/neurips-2024-statistics-datasets-benchmarks-track/), our submission seems to be on the verge of meeting the criteria, achieving an average score of 6.5 compared to the previous acceptance average of 6.6. We would be grateful if you could confirm that you have read our responses and let us know if there's anything we might have overlooked that could improve the score so that we can have the opportunity to showcase this work at the conference. We are eager to discuss any concerns that may remain open after our response. Thank you again for taking the time for this review.

---

> > > ### Comment · Reviewer_Tcpd · 2024-08-30
> > >
> > > Thanks for the comprehensive rebuttal. It confirms that my initial assessment was correct, and I believe that this paper is ready for publication.
> > >
> > > I would not worry unduly about the acceptance statistics; from prior experience as both a reviewer and an AC, the reported statistics seem heavily biased and, by design, do not account for withdrawn papers or papers without any rebuttal by the authors.

---

### Official Review · Reviewer_eDST · 2024-07-21
**Review for TSB-AD contribution**

**Rating:** 7
**Confidence:** 3
**Correctness:** The claims made in the submission are…
**Clarity:** The paper is well written.

**Review:**

The paper makes contributions by addressing some existing issues in time-series anomaly detection. The dataset integrity, metric reliability, and comprehensive benchmarking are well presented and systematically tackled.

However, the comparison with baseline results reported in other papers is missing, which might be potentially important to understand how far is it concerning the historically reported results on the un-curated datasets. The manual effort involved in curating the datasets and evaluating the algorithms is not detailed enough, making it a bit difficult to assess the quality of the conducted dataset curation.

While the mentioning of foundation models' data contamination is interesting, it needs more thorough examination by for example validating on unseen synthetic/industrial dataset.

**Strengths:**

- The paper provides a curated dataset that potentially enhances the resources available for time-series anomaly detection.

- It addresses biases in existing evaluation metrics, offering a seemingly more reliable framework for performance assessment.

- The benchmarking of 35 algorithms on curated dataset, including statistical methods and foundation models, offers valuable insights into their comparative performance.

- The findings challenge conventional wisdom, highlighting the effectiveness of simpler statistical methods over complex neural networks.

- The open-sourcing of the dataset and implementation promotes transparency and further research.

**Additional Feedback:**

N/A

**Documentation:**

The paper provides sufficient detail to support reproducibility, but more information on the manual effort involved in the curation process may be needed.

**Limitations:**

The paper mentioned a few limitation in it discussion.

**Opportunities For Improvement:**

- The comparison with baseline results reported in other papers is missing, which might be potentially important to understand how far is it concerning the historically reported results on the un-curated datasets.

- The manual effort involved in curating the datasets and evaluating the algorithms is not detailed enough, making it a bit difficult to assess the quality of the conducted dataset curation.

- While the mentioning of foundation models' data contamination is interesting, it needs more thorough examination by for example validating on unseen synthetic/industrial dataset.

**Relation To Prior Work:**

The discussion of how this work differs from previous contributions is clear, but it would benefit from a more explicit comparison with baseline results from their original publications.

**Summary And Contributions:**

The paper introduces TSB-AD, a new benchmark for time-series anomaly detection, attempting to address issues in existing datasets, evaluation metrics, and benchmarking practices. It provides a curated dataset from 33 domains, incorporates a set of metrics, and evaluates 35 detection algorithms. The findings suggest that simpler statistical methods often outperform advanced neural network architectures, highlighting the need for careful consideration of data contamination when using foundation models.

---

> ### Author Rebuttal · Authors · 2024-08-16
>
> We appreciate the effort the reviewer has taken to analyze our paper. The feedback is instrumental in guiding our revisions. Here is the response to your suggestions.
>
> > (1) Comparison with baseline results reported in other papers.
>
> |        | NAB  | NAB* | IOPS | IOPS* |
> |--------|------|------|------|-------|
> | NORMA [1]  | **0.67** | 0.58 | 0.55 | **0.76**  |
> | PCA [2]   | 0.63 | **0.69** | **0.77** | 0.74  |
>
> **Table 1:** The comparison of AUC-ROC results obtained from TSB-AD and other papers. \* indicates results from another paper with uncurated datasets [3].
>
> By analyzing the AUC-ROC results from our TSB-AD and those reported in other studies (as shown in **Table 1**), we identify significant changes in the performance of different methodologies. For example, PCA outperforms NORMA on the uncurated NAB dataset, whereas NORMA is more effective on the curated version. Conversely, NORMA excels on the IOPS* dataset but underperforms on the IOPS dataset. To enhance the reliability of those findings, given that current reports in various papers may have been based on different subsets of the datasets under different evaluation metrics, we will conduct further analyses on both pre- and post-curated datasets within TSB-AD. This approach will ensure a more fair and accurate comparison to understand how concerning are previously reported results in the community. Thanks for this excellent suggestion; we will provide a detailed comparative analysis in the revised manuscript.
>
> > (2) Details on manual effort in the dataset curating process.
>
> The primary objective of the dataset pruning process is to improve the overall quality of anomaly labeling. Within this task, three human annotators carry out the manual inspection process, who reach conclusions through consensus. This process involves `two` key steps:
> * First, annotators perform a visual inspection of the time series collection to identify prevalent flaws. These include (i) mislabeling, where similar patterns are inconsistently classified—some designated as anomalies and others not, (ii) bias cases, such as run-to-failure bias and single anomaly bias, and (iii) cases where datasets are unsuitable for the anomaly detection task due to a lack of context or unrealistic anomaly ratios, as illustrated in **Figure 1**. Annotators identify and remove time series that exhibit such flaws.
> * Second, for instances where the complexity of labeled anomalies exceeds the capabilities of human annotators, they proceed to the second step of the labeling quality assessment (Section 4.1.2). With the assistance of predictions from multiple anomaly detectors, human annotators are required to distinguish among several scenarios based on the flowchart depicted in **Figure 4**: (i) good label quality, where at least one anomaly detector successfully identifies the anomalies; (ii) bias within the dataset, which can be addressed by segmenting highly confident regions or extending the label; (iii) good label quality but the anomaly is inherently difficult to detect and (iv) the lack of sufficient in-context data to detect an anomaly.
>
> We have documented the rationale and procedures for each time series that is either removed or revised to improve label quality. These details will be made available in the official dataset release. Additionally, we will clarify this process further in Section 4.1 of our manuscript.
>
> > (3) Examination of data contamination problem in foundation models.
>
> Identifying time series that a foundation model has not encountered during pretraining is challenging due to the extensive coverage of publicly available time series in their pretraining data and sometimes ambiguous data source usage. Nevertheless, the **dedicated evaluation subset**, utilized for assessing the performance of a foundation model as detailed in their original publication, serves as a reliable source of time series that the model has not previously encountered. This allows us to utilize the dedicated evaluation subset for the effective analysis of data contamination problems. For instance, the MOMENT model [4] provides an example of such a case. Our analysis of the zero-shot (ZS) and fine-tuned (FT) versions of MOMENT [4]—where fine-tuning uses the initial segments of the time series as training data—on TSB-AD and the dedicated evaluation subset (denoted as Eval), as illustrated in **Table 2** below, indicates a significant performance decline on Eval, particularly for the zero-shot version of MOMENT. Instead, statistical/data mining methods (e.g., NORMA [1]) do not suffer from such problems and exhibit competitive performance in both settings. This underscores a critical issue of data contamination. The revised manuscript will provide a more comprehensive analysis to highlight this concern further.
>
> |            | MOMENT (ZS) | MOMENT (FT) |  NORMA  |
> |------------|-------------|-------------|---------|
> | TSB-AD     | 0.40        | 0.48        |  0.46   |
> | Eval       | 0.08        | 0.20        |  0.44   |
>
> **Table 2:** Comparative VUS-PR analysis between MOMENT and NORMA on TSB-AD and the dedicated evaluation subset (Eval).
>
> -------
> **Reference:**
>
> [1] P. Boniol et al. Unsupervised and scalable subsequence anomaly detection in large data series. The VLDB Journal, pages 1–23, 2021.
>
> [2] C. C. Aggarwal. Outlier Analysis. Springer International Publishing, 2 edition, 2017.
>
> [3] J. Paparrizos et al. Tsb-uad: an end-to-end benchmark suite for univariate time-series anomaly detection. Proceedings of the VLDB Endowment, 15(8):1697–1711, 2022.
>
> [4] M. Goswami et al. Moment: A family of open time-series foundation models. In ICML 2024.

---

> > ### Author Rebuttal · Authors · 2024-08-26
> >
> > Dear reviewer,
> >
> > We really appreciate your efforts during this tight review timeline and your recognition of our paper's contributions to addressing existing issues in time-series anomaly detection. In our previous response, we have tackled your concerns about the comparison with baseline results from other studies, the manual effort involved in the dataset curating process, and the examination of data contamination issues in foundation models.
> >
> > We are really happy and respect your current rating for our work. However, based on the acceptance statistics from recent years (https://papercopilot.com/statistics/neurips-statistics/neurips-2024-statistics-datasets-benchmarks-track/), our submission seems to be on the verge of meeting the criteria, achieving an average score of 6.5 compared to the previous acceptance average of 6.6. We would be grateful if you could confirm that you have read our responses and let us know if there's anything we might have overlooked that could improve the score so that we can have the opportunity to showcase this work at the conference. We are eager to discuss any concerns that may remain open after our response. Thank you again for taking the time for this review.

---

### Official Review · Reviewer_HHcx · 2024-07-25

**Rating:** 7
**Confidence:** 4
**Correctness:** Yes
**Clarity:** Yes

**Review:**

pros:
- The paper provides a detailed analysis of the issues with existing time series anomaly detection benchmarks from three perspectives and offers solutions (from my point of view, especially the **Point Adjustment** issue that remains prevalent in the current time-series AD study)
- A meticulous analysis of the problems with existing datasets such as the mislabeling issues, bias in datasets, and feasibility of datasets for anomaly detection
- Reasonable pipeline for each aspect. For example, the pipeline illustrated in the flowchart of Figure 4 for assessing label quality and the selection processes of evaluation metrics demonstrated in Figure 5.
- Thoroughly experimental results, especially involving the time-series foundation models.

cons (also the Opportunities For Improvement):
- I'm not quite clear on the basis for the approach mentioned in Section 4.1.1, where some univariate datasets are converted from multivariate time series datasets. Additionally, would the selection of top 40% of ranked time-series make the detection difficulty of the dataset overly simplistic?
- I'm also interested in the detection performance of other Large Language Models (LLMs), such as the one-fits-all model and the LLM4TS model, in the TSB-AD benchmark.
- Based on the experimental results discussed in Section 5.2,  What scenarios should Large Language Models (LLMs), RNN/CNN, or traditional machine learning/statistical models be used in?

Reference

[1] Zhou, Tian, et al. "One fits all: Power general time series analysis by pretrained lm." Advances in neural information processing systems 36 (2023): 43322-43355.

[2] Chang, Ching, et al. "Llm4ts: Aligning pre-trained llms as data-efficient time-series forecasters." arXiv preprint arXiv:2308.08469 (2024).

**Strengths:**

To avoid repetition, please refer to the above reviews.

**Additional Feedback:**

Null

**Documentation:**

Yes

**Limitations:**

To avoid repetition, please refer to the above reviews.

**Opportunities For Improvement:**

To avoid repetition, please refer to the above reviews.

**Relation To Prior Work:**

Yes

**Summary And Contributions:**

This paper proposed the TSB-AD benchmark for time-series anomaly detection tasks. TSB-AD tackles three aspects in tims-series AD scenarios, including Data Integrity, Metric Reliability, and Comprehensive Benchmarking. TSB-AD is open-sourced and implemented in github.

---

> ### Author Rebuttal · Authors · 2024-08-16
>
> We are thankful for the reviewer's insightful comments. Please find our responses below:
>
> > (1) Basis for converting multivariate time series into univariate and threshold selection.
>
> The primary objective of converting multivariate time series into univariate formats is to enhance the **diversity** and **size** of our dataset collection. This strategy is underpinned by the following observations:
>
> * In numerous multivariate datasets, only a limited number of channels (often just one) provide valuable information for anomaly detection, while other channels contain categorical, binary, or random values.
> * Our correlation analysis, which evaluates the relationship between the anomaly score of each channel and the ground truth anomaly labels, demonstrates that certain channels exhibit a stronger correlation with the ground truth than others.
>
> These observations helped us transform the informative channels of multivariate time series into univariate time series datasets while ensuring the ignored channels do not contribute to the detection of anomalies.
>
> The selection threshold of the top 40% for each evaluation measure is determined by balancing the need to maximize the diversity of time series with the practical considerations of subsequent processing efforts. A strict threshold may result in overlooking potentially valuable cases. Instead, by using a relatively broader threshold, we ensure that strong candidates will not be overlooked. It is crucial to note that the dataset pruning process is **iterative**; any time series with suboptimal labeling that passes initial stages can be addressed and removed in subsequent iterations. Therefore, it's beneficial to be initially less strict regarding the threshold. Following the conversion of multivariate datasets to univariate formats, we have successfully incorporated 7 additional univariate datasets into our collection. We will clarify this procedure further and refine our manuscript accordingly.
>
> > (2) The detection performance of LLMs-based methods.
>
> We are more than happy to periodically incorporate additional LLM-based methods into TSB-AD. In particular, we plan to build timeseries.org into a portal containing a frequently updated leaderboard.
>
> Regarding the suggested methods, the LLM4TS model results will be included once the code is publicly available (currently it's not), and the results for the one-fits-all model (OFA) have already been provided in Appendix D, achieving a VUS-PR of `0.22` compared to the top-performing time-series foundation model, MOMENT, achieving `0.48`.
>
> > (3) Application scenarios of different kinds of anomaly detectors.
>
> The key takeaway for users on selecting anomaly detectors is as follows: statistical models are recommended for time series exhibiting periodic patterns, subsequence anomalies, or a higher anomaly ratio, whereas deep-learning-based methods, including RNNs/CNNs and LLMs, are preferable for scenarios with point-based anomalies, such as those seen in stock market (Stock dataset) and web services (IOPS dataset), or situations with abundant training data.
>
> Detailed experimental analyses covering various anomaly types and domains are provided in Appendix D due to space limitations in the main text. We will summarize and integrate the findings into section 5.2 to address this important comment.

---

> > ### Author Rebuttal · Authors · 2024-08-26
> >
> > Dear reviewer,
> >
> > We really appreciate your efforts during this tight review timeline and the recognition of the strengths of our paper, including its detailed analysis, comprehensive experimental results, and the reasonable pipeline for each aspect. In our previous response, we have addressed your concerns regarding the basis for converting multivariate time series, the detection performance of LLMs-based methods, and the application scenarios for various anomaly detectors.
> >
> > We are really happy and respect your current rating for our work. However, based on the acceptance statistics from recent years (https://papercopilot.com/statistics/neurips-statistics/neurips-2024-statistics-datasets-benchmarks-track/), our submission seems to be on the verge of meeting the criteria, achieving an average score of 6.5 compared to the previous acceptance average of 6.6. We would be grateful if you could confirm that you have read our responses and let us know if there's anything we might have overlooked that could improve the score so that we can have the opportunity to showcase this work at the conference. We are eager to discuss any concerns that may remain open after our response. Thank you again for taking the time for this review.

---

### Author Rebuttal · Authors · 2024-08-16

We sincerely appreciate the time and effort the reviewers have dedicated to providing detailed comments for our manuscript. We thank reviewers for the recognition of the strengths of our paper:
* Reliable benchmark and strong contribution to the community (Reviewer HHcx, eDST, Tcpd)
* Comprehensive and thorough analysis (Reviewer HHcx, eDST, Tcpd)
* Well-written and structured draft (Reviewer eDST, Tcpd, SW7e)

Below, we summarize the common feedback and concerns and provide our responses. Please refer to our individual responses for specific comments related to each reviewer.

* The Role of Manual Effort in Dataset Pruning.

The primary objective of the dataset pruning process is to improve the overall quality of anomaly labeling. This iterative process begins with a large collection of heterogeneous time series. The two-step manual inspection process comprises (i) identifying common flaws such as mislabeling, biases, and the feasibility of datasets for anomaly detection, as illustrated in Figure 1; (ii) conducting algorithmic tests to evaluate label quality and distinguish between good labels, poor labels, and cases of difficult anomalies, as depicted in the flowchart in Figure 4. The final decision to remove or revise the time series is made based on the consensus of 3 human annotators.

* Comparison with Other Studies.

Our benchmark addresses significant yet often overlooked issues in the time-series anomaly detection literature, including flawed datasets, biased evaluation metrics, and inconsistent benchmarking practices. Our work consolidates scattered insights from various communities into a coherent practice and builds upon the accumulated knowledge of the field to establish a reliable benchmark for time-series anomaly detection. For this purpose, we have created the first large-scale manually annotated dataset in this field, and our evaluation benchmark includes the union of best solutions from prior studies and the integration of foundation models for the first time in time-series anomaly detection benchmarks. Utilizing our identified reliable evaluation metrics, we reveal new findings about top-performing methodologies, previously overlooked methods, and insights into foundation models and neural-network-based methods. We hope the benchmark will advance the community and provide practical guidance for practitioners.

* Details on Experimental Setup and Analysis.

Due to space constraints in the main text, we have included a comprehensive description of the experimental settings in Appendix C, which covers implementation details and hyperparameter tuning. Additionally, further experimental results, including analyses of different types of anomalies and case studies, are detailed in Appendix D.

Moving forward, we will (i) review the entire manuscript to address language issues and enhance detail in response to reviewer feedback; (ii) release our datasets on Zenodo and a new portal on timeseries.org, along with a 'pip install' version of TSB-AD to facilitate straightforward utilization of our benchmark suite; (iii) continually update the benchmark with emerging datasets and methodologies, and host an online leaderboard on timeseries.org.

---

### Comment · Area_Chair_54tm · 2024-08-20
**Please response to the rebuttal**

Dear Reviewers,

Thank you so much for your efforts! In the current period, please check the response and discuss it with the authors if needed.

Thanks again,

Your AC

---

> ### Comment · Reviewer_Tcpd · 2024-08-30
>
> Thanks for checking in! All my concerns have been addressed, and I maintain that this paper is ready for publication.

---

### Decision · Program_Chairs · 2024-09-26

**Decision:**

Accept (Poster)

**Comment:**

This paper presents the TSB-AD benchmark for time-series anomaly detection tasks. It addresses the limitations of existing time-series anomaly detection datasets, metrics, and comparisons, and introduces a comprehensive benchmark. The paper provides a curated dataset from 33 diverse domains and compares 35 detection algorithms on the datasets. The amount of work is significant, and the reviewers are generally very positive about the contributions to the community. The rebuttal sufficiently addresses most of the comments. Please make the dataset more easily accessible by creating a dedicated website, a pip-installable package, and a leaderboard as promised in the rebuttal.